# Enhanced chiroptic properties of nanocomposites of achiral plasmonic nanoparticles decorated with chiral dye-loaded micelles

Tonghan Zhao [1,4], Dejing Meng[2,4], Zhijian Hu[2], Wenjing Sun[1], Yinglu Ji[2], Jianlei Han[1], Xue Jin[1], Xiaochun Wu[2,3] ✉ & Pengfei Duan [1,3] ✉

The development of circularly polarized luminescence (CPL)-active materials with both large luminescence dissymmetry factor ($g_{lum}$) and high emission efficiency continues to be a major challenge. Here, we present an approach to improve the overall CPL performance by integrating triplet-triplet annihilation-based photon upconversion (TTA-UC) with localized surface plasmon resonance. Dye-loaded chiral micelles possessing TTA-UC ability are designed and attached on the surface of achiral gold nanorods (AuNRs). The longitudinal and transversal resonance peaks of AuNRs overlap with the absorption and emission of dye-loaded chiral micelles, respectively. Typically, 43-fold amplification of $g_{lum}$ value accompanied by 3-fold enhancement of upconversion are obtained simultaneously when Au@Ag nanorods are employed in the composites. More importantly, transient absorption spectra reveal a fast accumulation of spin-polarized triplet excitons in the composites. Therefore, the enhancement of chirality-induced spin polarization should be in charge of the amplification of $g_{lum}$ value. Our design strategy suggests that combining plasmonic nanomaterials with chiral organic materials could aid in the development of chiroptical nanomaterials.

Circularly polarized luminescence (CPL)-active materials have significant applications in various fields, ranging from 3D display to optical storage and information encryption[1–4], as well as asymmetric synthesis[5,6]. The luminescence dissymmetry factor ($g_{lum}$) is employed to evaluate the circular polarization degree of CPL. This value is determined by $g_{lum} = 2 \times (I_L - I_R)/(I_L + I_R)$, where $I_L$ and $I_R$ denote the intensities of left- and right-handed circularly polarized light, respectively[7]. There is a strong need for CPL-active substances with a high level of circular polarization and luminescence efficiency. Up to now, simple organic molecules (SOMs) with highly efficient emission have been developed into considerable candidates for fabricating CPL-active materials[8,9]. However, currently, CPL-active SOMs are incapable of high $g_{lum}$ value[10]. Various approaches have been devoted to enhancing the $g_{lum}$ value, including supramolecular self-assembly[11,12], aggregation-induced emission[13,14], chiral liquid crystal[15–18], fluorescence resonance energy transfer[19–23], and triplet-triplet annihilation-based photon upconversion (TTA-UC)[24–26]. The last TTA-UC system has drawn lots of attention due to its ability to absorb low-frequency light to generate high-frequency CPL[27–31], i.e., the upconverted CPL (UC-CPL). Moreover, the obtained UC-CPL has wide potential applications

[1]CAS Key Laboratory of Nanosystem and Hierarchical Fabrication, National Center for Nanoscience and Technology (NCNST), No.11, ZhongGuanCun BeiYi-Tiao, Beijing 100190, P. R. China. [2]CAS Key Laboratory of Standardization and Measurement for Nanotechnology, National Center for Nanoscience and Technology (NCNST), No.11, ZhongGuanCun BeiYiTiao, Beijing 100190, P. R. China. [3]University of Chinese Academy of Sciences, Beijing 100049, P. R. China. [4]These authors contributed equally: Tonghan Zhao, Dejing Meng. ✉e-mail: wuxc@nanoctr.cn; duanpf@nanoctr.cn

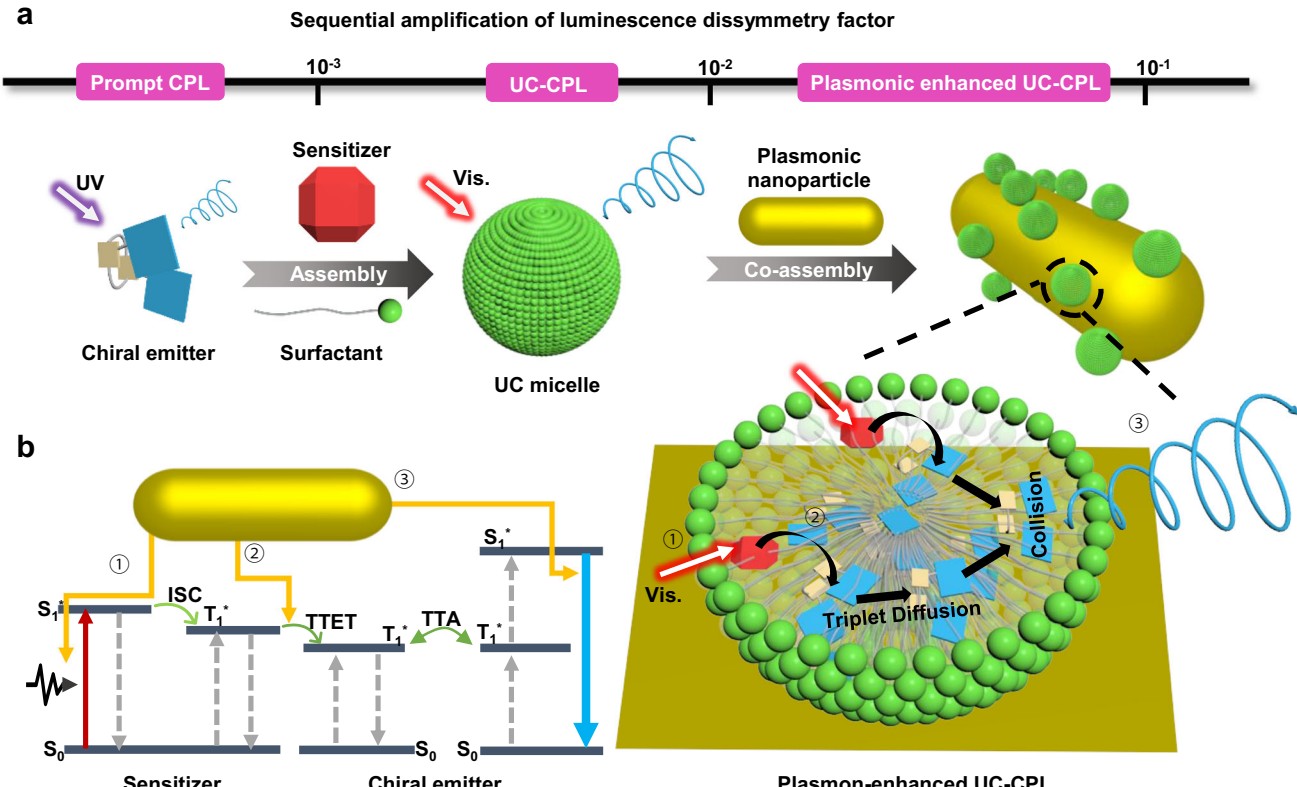

**Fig. 1 | Concurrently amplified dissymmetry factor $g_{lum}$ and upconversion emission from plasmon-micelle composites in aqueous solution. a** Under the excitation of UV light, the chiral emitter shows prompt CPL with a $g_{lum}$ value around $1.0 \times 10^{-3}$. When chiral emitter/sensitizer/CTAB micelles assemble into upconverted micelles, the high-frequency UC-CPL can be obtained under excitation with low-frequency light. The upconverted micelles exhibit amplified $g_{lum}$ values between $1.0 \times 10^{-3}$ and $1.0 \times 10^{-2}$. Furthermore, plasmon-micelle composites that are formed by upconverted micelles and plasmon nanoparticles through electrostatic interaction can produce plasmon-enhanced UC-CPL, in which upconverted emission intensity and $g_{lum}$ value ($>1.0 \times 10^{-2}$) are enhanced simultaneously. **b** Illustration of multiple photophysical processes in plasmon-assisted TTA-UC system. Source data are provided as a Source Data file.

in enantioselective photopolymerization and multi-level optical encryption[26,32–34]. Although TTA-UC has been demonstrated as an efficient strategy for enhancing CPL performance, the amplified $g_{lum}$ value in dilute solution still doesn't meet the requirement of practical applications. Meanwhile, the absolute emission intensity is suppressed by exciton-exciton annihilation. Therefore, there is still an intense demand for further amplifying $g_{lum}$ value and simultaneously improving upconversion intensity.

Metallic nanostructures support localized surface plasmon resonance (LSPR), which results in the strong enhancement of electromagnetic field at the surface of these nanostructures[35–38]. Due to the strong and tunable LSPR features, metallic nanoparticles have attracted extensive research interests and offer potential applications in surface-enhanced Raman scattering[39,40], plasmonic circular dichroism[41–46], and surface-enhanced luminescence[47,48]. To date, the LSPR has also successfully enhanced the TTA-UC emission[49,50]. It has been demonstrated that LSPR can lead to a beneficial photoexcitation enhancement of the sensitizers and acceleration in the radiative decay process of the emitters[51]. Nevertheless, LSPR-enhanced CPL has never been involved in TTA-UC systems. In 2014, Harada and his co-workers reported the plasmon-enhanced CPL by silver nanoparticles in chiral surfactant assemblies[52]. Shi et al. demonstrated LSPR-boosted CPL of europium polyoxometalates[53]. Recently, silver nanowires and gold triangular nanoprisms have also been employed to amplify the circular polarization of chiral emitters[54,55]. However, in those systems, the plasmon-enhanced CPL was generated directly under the excitation of high-frequency light. As for TTA-based UC-CPL systems, the effect of LSPR on upconverted CPL can be complicated since TTA-UC involves multiple photophysical processes. As

sketched in Fig. 1b, the LSPR of metal nanoparticles can probably influence a cascade of events in UC-CPL: (1) the photons absorption of sensitizers, (2) the triplet dynamics: transfer, diffusion, collision, and annihilation, (3) the radiative transition of chiral emitters. In this case, the energy-level matching between TTA-UC sensitizer/annihilator pairs and plasmonic nanostructures requires a considerate design.

Herein, we report a plasmonic noble metal nanoparticle-assisted chiral upconversion system which can obtain the concurrent enhancement of upconversion emission and $g_{lum}$ (Fig. 1a). Under the excitation of UV light, the micellar aggregate (*R*-1M) consisting of hydrophobic chiral emitter *R*-1 (molecular structure in Fig. 2a) and hexadecyltrimethyl ammonium bromide (CTAB) exhibited prompt CPL with a $g_{lum}$ value around $1.5 \times 10^{-3}$. After assembling with sensitizer Pd(II) meso-tetraphenyl tetrabenzoporphine (PdTPBP), the upconverted micellar aggregates (*R*-1UCM) could emit upconverted circularly polarized light upon excitation with 635 nm laser. Moreover, the $g_{lum}$ value was magnified more than four times and up to $6.4 \times 10^{-3}$. After mixing with polystyrene sulfonate (PSS)-modified gold nanorods (AuNRs), the upconverted micelles could adhere to the surface of AuNRs through electrostatic interaction. It should be noted that the plasmon-micellar composites (*R*-1UCM/AuNRs) exhibited strong amplification of UC-CPL. When the molar ratio of AuNRs and *R*-1 was $3.0 \times 10^{-6}/1$, the $g_{lum}$ value increased to $4.6 \times 10^{-2}$. At the same time, the upconverted emission intensity of *R*-1UCMP/AuNRs was approximately two times larger than that of the pristine upconverted micelles. Based on the control experiments with different types of plasmonic particles and the finite-difference time-domain (FDTD) calculations, as well as the measurement of time-resolved upconversion emission

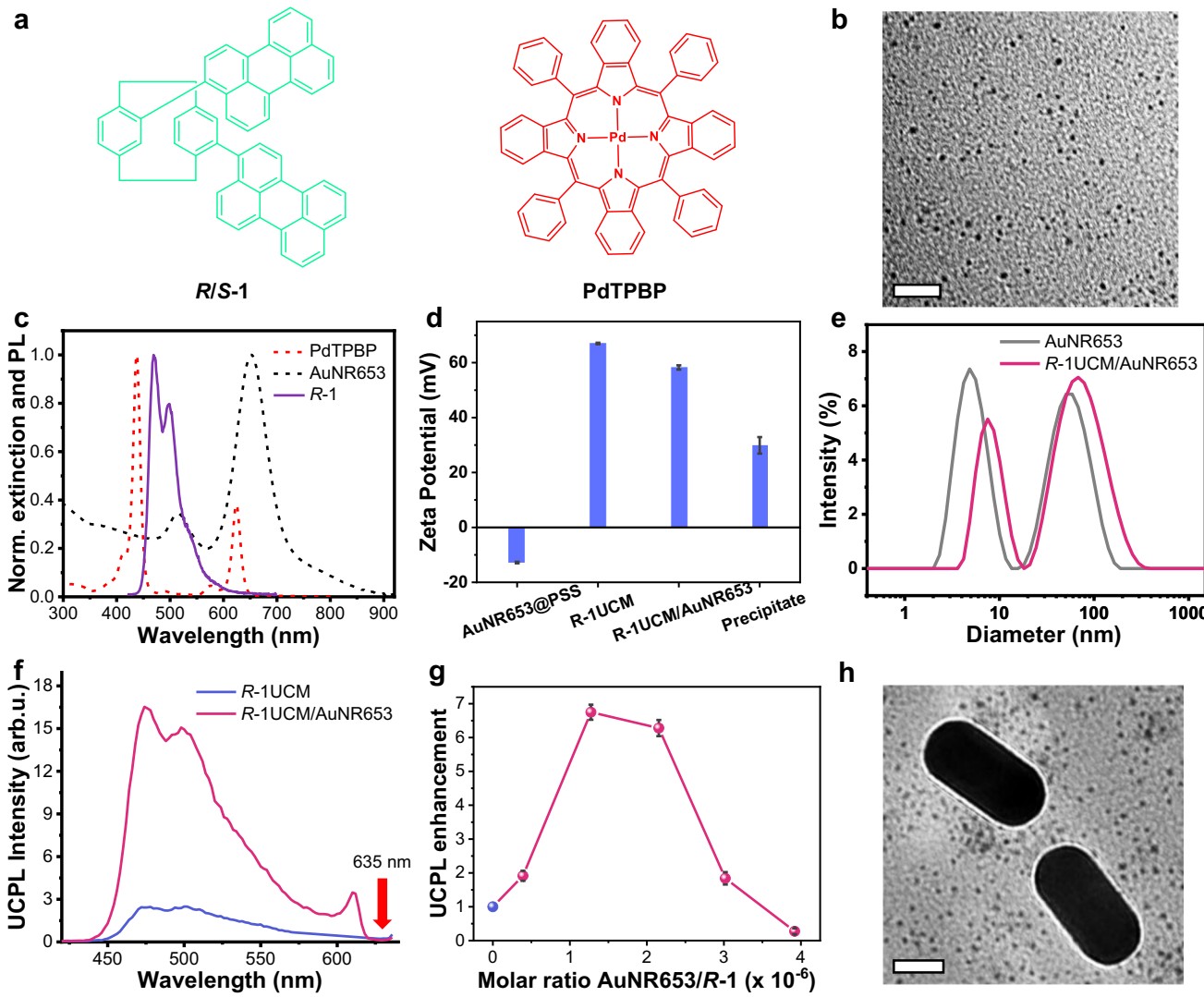

**Fig. 2 | Photon upconversion of plasmon-micelle composites. a** Molecular structures of chiral emitters *R/S*−1 and sensitizer PdTPBP. **b** TEM image of *R*−1UCM. Scale bar = 20 nm. **c** Normalized extinction (dash line) and photoluminescence (PL, solid line) spectra of PdTPBP and *R*−1 in THF and AuNR653 in water. $\lambda_{ex}$ = 400 nm. [*R*-1] = [PdTPBP] = $10^{-5}$ mol $L^{-1}$, [AuNR653] = $10^{-10}$ mol $L^{-1}$. **d** Zeta potential of *R*-1UCM, *R*-1UCM/AuNR653 and AuNR653 in water. All error bars show mean ± standard deviation. *n* = 3 independent experiments. **e** Particle size distribution for a stable dispersion of AuNR653 and *R*-1UCM/AuNR653 in water obtained by DLS. **f** Upconverted PL spectra of *R*-1UCM and *R*-1UCM/AuNR653 in deaerated water

under excitation of 635 nm laser. The measured UC luminescence of the *R*-1UCM/AuNR653 composite was 6.7-fold that of *R*-1UCM at 476 nm. [*R*-1] = $5 \times 10^{-5}$ mol $L^{-1}$, [PdTPBP] = $10^{-5}$ mol $L^{-1}$, [CTAB] = $10^{-2}$ mol $L^{-1}$, molar ratio AuNR653/*R*-1 = $1.3 \times 10^{-6}$/1. A 635 nm short-pass filter was set in front of the detector to remove the scattering incident light. **g** Upconverted intensity ratio between *R*-1UCM and *R*-1UCM/AuNR653. The blue sphere is the *R*-1UCM only. The power density of the 635 nm laser was 2346 mW $cm^{-2}$. All error bars show mean ± standard deviation. *n* = 3 independent experiments. **h** TEM image of *R*−1UCMP/AuNR653 composites. Scale bar = 20 nm. Source data are provided as a Source Data file.

spectra, we verified the mechanism that LSPR can enhance the absorption of sensitizers and accelerate the radiative decay of the chiral emitters. Although the effect of LSPR on TTET is not entirely clear, through transient absorption spectra, we observed that the triplet excitons of *R*-1 (spin-polarized triplet excitons[25]) accumulated fast in the presence of AuNRs. Therefore, the enhancement of chirality-induced spin polarization should be a significant reason for amplifying the circular polarization of UC-CPL. It's a general way to modulate the upconversion process of dye molecules by utilizing the local electromagnetic field of metal nanoparticles, and it will give more insights into the development of the optoelectronics field.

## Results

### *R/S*-1 and CTAB assembled into chiral micelles

Planar chirality-based *R*-1 and *S*-1 with perylene chromophores (Fig. 2a) were CPL-active emitters[25]. Therefore, we utilized *R/S*-1 and CTAB to

fabricate CPL-active micellar aggregates (*R/S*-1M) through the co-assembly method. After fixing the concentration of CTAB ($10^{-2}$ mol $L^{-1}$) in water, a series of micellar aggregates with various concentrations of *R*-1 were investigated. The optimized concentration of *R*-1 was $5 \times 10^{-5}$ mol $L^{-1}$ because of the strongest emission intensity (Supplementary Fig. 1). Although the quantum yield of *R*-1M was lower than *R*-1 in THF, it had no severe bathochromic shift and aggregation-caused emission quenching in fluorescence in comparison to directly dispersed *R*-1 in water (Supplementary Table 1 and Fig. 2). Those results suggested that *R*-1 were well stabilized in the presence of micelles. It is well-known that energy transfer can occur between sensitizer PdTPBP and perylene derivatives in TTA-UC[56]. Therefore, we further introduced PdTPBP to co-assemble with *R*-1 and CTAB to get *R*-1UCM. Transmission electron microscopy (TEM) images showed that *R*-1UCM were nanoscale with an average diameter of 4 nm (Fig. 2b). In addition, upconversion spectra of *R*-1UCM with different incident light power

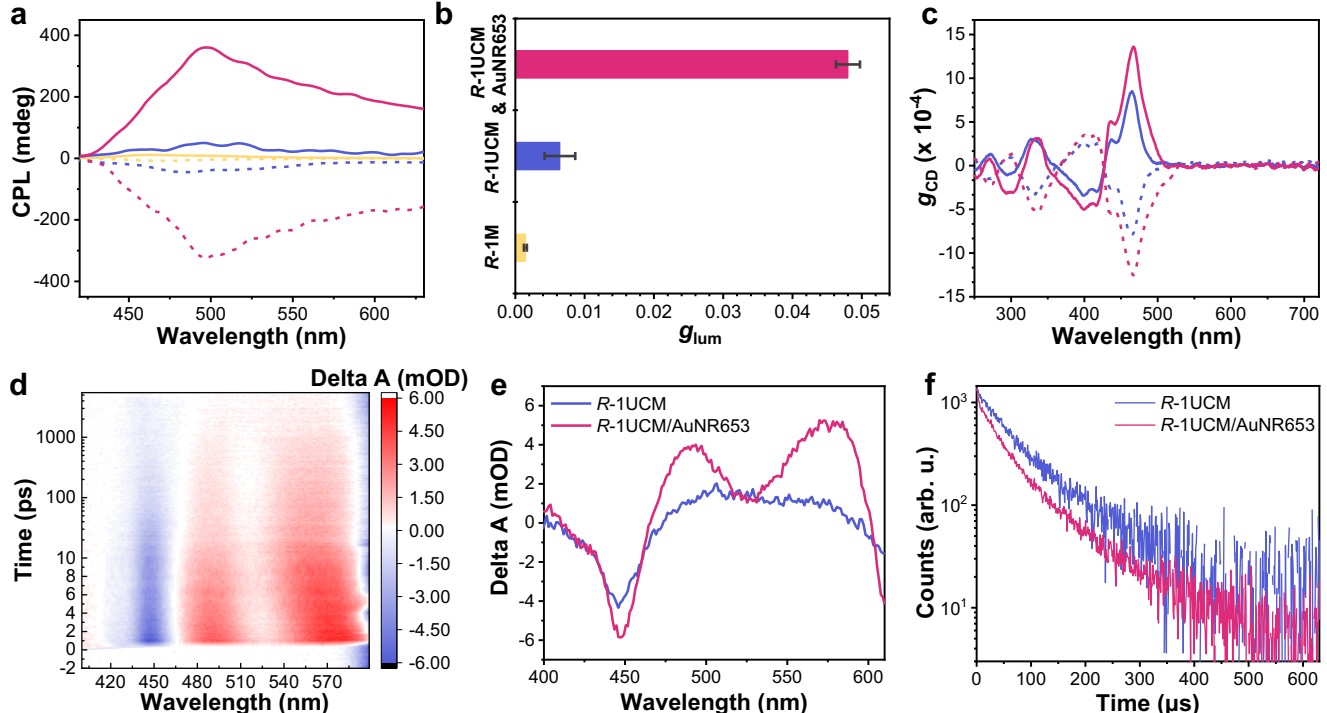

**Fig. 3 | Chiroptical properties of plasmon-micelle composites. a** CPL spectra of $R$−1M (yellow line) and $S$−1M (yellow dash line) in water excited by 400 nm. UC-CPL spectra of $R$−1UCM (blue line), $S$−1UCM (blue dash line), $R$−1UCM/AuNR653 (red line), and $S$−1UCM/AuNR653 (red dash line) in deaerated water excited by 635 nm laser. **b** Dissymmetry factor $g_{lum}$ values of CPL, UC-CPL, and plasmon-enhanced UC-CPL at peak values of emission. All error bars show mean ± standard deviation. $n = 3$ independent experiments. **c** CD dissymmetry factor $g_{CD}$ of $R$-1M (blue line), $S$-1M (blue dash line), $R$-1M/AuNR653 (red line), and $S$-1M/AuNR653 (red dash line) versus wavelength. **d** Two-dimensional transient absorption spectra of $R$-1UCM/AuNR653 in deoxygenated water. **e** Transient absorption spectra of $R$-1UCM and $R$-1UCM/AuNR653 in deoxygenated water after excitation at 635 nm, delay time: 1 ps. **f** Upconversion decay at 476 nm of $R$-1UCM and $R$-1UCM/AuNR653. [$R$-1] = $5 \times 10^{-5}$ mol L$^{-1}$, [PdTPBP] = $10^{-5}$ mol L$^{-1}$, [CTAB] = $10^{-2}$ mol L$^{-1}$, molar ratio AuNR653/$R$−1 = $3 \times 10^{-6}$/1. A 635 nm short-pass filter was used. The power density of the 635 nm laser was 2000 mW cm$^{-2}$. Source data are provided as a Source Data file.

density of 635 nm were recorded (Supplementary Fig. 3c). A threshold $I_{th}$ of 1637 mW cm$^{-2}$ was obtained, above which TTA becomes the main triplet deactivation channel for the chiral emitters (Supplementary Fig. 3d).

### Plasmon-enhanced photon upconversion of $R/S$-1UCM
Considering that the $R$-1UCM were formed by cationic surfactants, the as-prepared AuNRs with a longitudinal LSPR at 653 nm (AuNR653) were coated with PSS and thus gave negatively charged nanorods. After the attaching of PSS, the Zeta potential of AuNR653 inverted from +50 to −12 mV (Fig. 2d). The opposite surface charges between $R$-1UCM and AuNRs allow to fabricate plasmon-micelle composite through electrostatic interaction. As shown in Fig. 2c, the longitudinal LSPR band of the AuNR653 overlapped with the absorption band of PdTPBP, meanwhile, its transverse LSPR matched with the fluorescence band of $R$-1. This suggested a potential optical coupling between AuNR653 and the upconversion pairs. Based on the above results, we mixed AuNR653 and $R$-1UCM in an aqueous solution. The mixed solution was kept undisturbed for 10 h, and the Zeta potential decreased from +67 to +58 mV when AuNR653 was added into the $R$-1UCMP solution (Fig. 2d). To further identify the successful formation of $R$-1UCM/AuNR653 hybrids, the precipitates were collected after centrifugation at 6797×$g$ for 10 min and redispersed with water. The Zeta potential of the composites was +30 mV, while a pure AuNR653 was −12 mV. Moreover, in $R$-1UCM/AuNR653 system, a growth of hydrodynamic diameters of AuNR653 from 4.9 and 58.8 nm to 7.5 and 68.1 nm was studied by dynamic light scattering (Fig. 2e). Therefore, the electrostatic attraction enabled attaching of the positively charged $R$-1UCM to the surface of the negatively charged AuNR653. Under the

excitation of a 635 nm laser, the measured upconverted luminescence of $R$-1UCM/AuNR653 composite was 6.7-fold that of $R$-1UCM at 476 nm (Fig. 2f). Meanwhile, the $I_{th}$ of $R$-1UCM/AuNR653 decreased to 1318 mW cm$^{-2}$ (Supplementary Fig. 3b). The upconversion quantum yield of $R$-1UCM increased from 0.2 to 0.5% after mixing with AuNR653 (Supplementary Fig. 3e). The above phenomenon revealed that the AuNR653 could enhance the luminescence of $R$-1UCM. We further investigated the relationship between the enhancement of photon upconversion and rod concentration. By increasing the AuNR653 amount, the upconverted luminescence intensity versus the concentration of metal nanoparticles showed a volcano-shape curve (Fig. 2g). When the ratio of AuNR653 and $R$-1UCM reached $3.9 \times 10^{-6}$, upconverted luminescence intensity would decrease due to the competing absorption at 635 nm between sensitizers and AuNR653 and the reabsorption of upconverted luminescence by AuNR653.

### Plasmon-enhanced chiroptical properties
Considering the planar chirality of $R/S$-1, we explored the chiroptical properties of $R/S$-1M assemblies. The mirror-imaged circular dichroism (CD) and CPL signals of $R/S$-1M were consistent with the $R/S$-1 in dilute solution (Fig. 3a and Supplementary Fig. 6), suggesting that the assemblies of chiral emitters and CTAB had little influence on chiroptical properties. The $g_{lum}$ values of prompt CPL at 481 nm were $+1.4 \times 10^{-3}$ and $-0.9 \times 10^{-3}$ for $R$-1M and $S$-1M, respectively. A positive UC-CPL curve was observed in $R$-1UCM upon being excited by a 635 nm laser. The corresponding $g_{lum}$ value of UC-CPL raised up to $+6.4 \times 10^{-3}$, presenting 4.3 times larger than the prompt CPL. The enhancement effect was additionally obtained in $S$-1UCM, which could be attributed to the chirality-induced electron spin polarization during the TTA-UC

process[25]. Furthermore, we thoroughly studied the plasmon-assisted UC-CPL in $R$-1UCM/AuNR653 system. With the addition of AuNR653, the $g_{lum}$ values of plasmon-enhanced UC-CPL gradually increased and reached a maximum when the molar ratio AuNR653/$R$-1 was $3.9 \times 10^{-6}$/1, which was 7.5 times and even 32.0 times larger in magnitude compared with the $g_{lum}$ values of $R$-1UCM and $R$-1M, respectively. As expected, the enantiomeric $S$-1UCM system showed the same enhancement phenomenon (Fig. 3a, b and Supplementary Fig. 4). The true enhancement factor per attached micelle should be larger because a fraction of the micelles was free, as exhibited in the TEM image of $R$-1UCM/AuNR653 (Fig. 2h). Here, we attempted to obtain the real enhancement factor through comparing the integrated extinction intensities of $R$-1UCM and the supernatant of $R$-1UCM/AuNR653 after removal of AuNR653 by centrifugation. In this operation, the $R$-1UCM/AuNR653 composite with a molar ratio AuNR653/$R$-1 at $3.0 \times 10^{-6}$/1 was selected due to high-concentration AuNR653 can quench the upconverted emission. As shown in Supplementary Fig. 5, the extinction intensity of the supernatant was 58% of the pure $R$-1UCM, which meant that approximately 42% of the micellar aggregates were combined with the AuNR653. Accordingly, the corrected enhancement factors were 3.3 and 15.7 times for upconverted emission and $g_{lum}$ value, respectively. With the deviations in the centrifugation process, this method was not perfect, but sufficient to roughly estimate the enhancement factor in a very simple and convenient fashion. Above all, the LSPR-assisted CPL enhancement method was successfully achieved in chiral organic-plasmon upconverted luminescence systems. Moreover, there was an obvious increase in ground-state chirality through measuring the CD signal (Supplementary Fig. 6) in both the $R$-1 and $S$-1 systems in the presence of AuNR653. To eliminate the influence of absorption, we converted those CD signals to the absorption dissymmetry factor ($g_{CD}$). As shown in Fig. 3c, the maximal $g_{CD}$ factor of plasmon-micelle composites at 466 nm was $1.4 \times 10^{-3}$, which was almost the twice larger than the one of individual micelle particles. These results clearly verified that either ground- or excited-state chirality can be enhanced by the plasmonic LSPR effect. We noted that the amplification level of CD was much smaller than that of UC-CPL. This may be explained by the extinction band of $R$-1 was not well matching the LSPR wavelength of AuNR653. Additionally, the measurement of regular downshifting CPL was carried out in $R$-1M/AuNR653 composites. Different from the UC-CPL, upon excitation with 400 nm, the $R$-1 mixed with different concentrations of AuNR653 only showed a slight increase in $g_{lum}$ values (Supplementary Fig. 7). This result indicated that although the emission band of $R$-1 overlapped with the transverse LSPR of AuNR653, the electromagnetic field of AuNR653 was too weak to allow a dramatic enhancement effect at such an excitation wavelength. This behavior will be further investigated in the following experiments by employing plasmonic nanorods with various longitudinal LSPR wavelengths.

## Mechanism of the enhancement

To gain further insights into the mechanism of LSPR-enhanced upconversion and CPL, we utilized transient absorption spectroscopy to dig into the behavior of excited triplet states. The comparison between PdTPBP and $R$-1UCM transient spectra were illustrated in Supplementary Fig. 8. Two ground-state bleaches (GSB) can be observed at 442 and 492 nm, which were in accordance with the singlet transition of PdTPBP. Furthermore, PdTPBP also showed excited-state absorption (ESA) at 484 and 524 nm due to $T_1 - T_n$ transitions[27]. In the $R$-1UCM system, the GSB signal shifted to 446 nm because of the presence of $R$-1. A signal appeared at 500 nm, which resulted from the $T_1 - T_n$ transitions of $R$-1. The transient absorption of $R$-1UCM/AuNR653 also exhibited a GSB signal at 446 nm, while two ESA peaks can be seen at 494 and 570 nm (Fig. 3d). After analyzing the transient absorption of AuNR653, we concluded that the split of triplet transitions peak of $R$-1 around 500 nm was caused by the GSB of AuNR653 at 520 nm.

Nevertheless, either at 446 or 500 nm, the transient absorption of $R$-1UCM/AuNR653 showed higher intensities than that of $R$-1UCM, demonstrating an LSPR enhancement effect for GSB and ESA (Fig. 3e and Supplementary Fig. 8d). In terms of exciton dynamics, the ESA decay of the $R$-1UCM and $R$-1UCM/AuNR653 were monitored at 446 and 500 nm, and the $R$-1UCM/AuNR653 showed faster decay at those peaks (Supplementary Fig. 9). Additionally, time-resolved photoluminescence spectra revealed that the lifetime of $R$-1UCM showed an obvious decrease from 102 to 82 µs in the presence of AuNR653 (Fig. 3f), which resulted from the LSPR-caused faster emission rates[49,57], further demonstrating the coupling of LSPR to $R$-1/PdTPBP pairs.

It's well-known that the LSPR features dramatically depend on the composition, shape, and size of the plasmonic nanostructures, three types of PSS-modified noble metal nanorods were synthesized to be involved in the hybrid systems. Two of them were gold nanorods with longitudinal LSPR bands at 737 and 812 nm, respectively. The rest was gold-core-silver-shell nanorods with a longitudinal LSPR band at 659 nm, and this core-shell nanoparticle was expected to have a stronger plasmonic response due to the presence of silver[37,58]. It was clear to see the different coupling degrees between plasmonic bands and excitation light due to the distinction of overlaps in the spectra (Fig. 4e). Moreover, the LSPR activities of these nanorods remained stable as their extinction spectra only showed a slight shift or broadening after the hybrid process (Supplementary Fig. 10). The upconversion emission of $R$-1UCM coupled with those Au and Au@Ag nanorods was investigated. The molar ratio of AuNR653 to $R$-1 was fixed at $3.0 \times 10^{-6}$/1. To avoid the influence of unequal adsorption of $R$-1UCM on the surface of metal nanoparticles, the concentrations of various metal nanorods were determined by their monomer's surface area, ensuring that the total surface area of those plasmonic particles were the same (Fig. 4a–d and Table 1). As shown in Fig. 4f, for gold nanorods, the increasing spectral overlap between the plasmon band and 635 nm laser would lead to the enhancement of upconversion intensity. Interestingly, the maximal upconversion enhancement was obtained in $R$-1UCM/AuNR@Ag659 system. To further explore the mechanism, we computed the electric field intensities of plasmonic nanoparticles by FDTD simulation. The calculated results suggested that the electric field was much stronger at the ends than the field at the side for the discrete AuNR when excited by a 635 nm laser, which was in line with the excitation of the sensitizers (Fig. 4g and Supplementary Fig. 11). Furthermore, the local electric fields on the metal surface decreased quickly when the longitudinal LSPR band gradually red-shifted and was far away from the 635 nm laser due to the increased aspect ratio of gold nanorods. After coating a thin Ag shell, AuNR@Ag659 showed 1.5-fold higher electric field enhancement compared to pure AuNR653, which further led to better-upconverted luminescence, agreeing with the observations in Fig. 4f. Those results demonstrated that the stronger electromagnetic field of noble metal nanorods, the more enhancement of photon upconversion. Additionally, Fig. 4h showed a plot of $g_{CD}$ against $g_{lum}$ for the results of chiral micelles and micelle/plasmon systems. Once chiral micelles were coupled with plasmonic nanoparticles, either $g_{CD}$ or $g_{lum}$ values would be enhanced compared with pure chiral micelle systems. The maximal $g_{CD}$ around 466 nm of AuNR653, AuNR737, and AuNR812 were comparable because of the similar electromagnetic field intensities at the transverse LSPR band around 520 nm. However, the enhancement of $g_{lum}$ values exhibited large differences with the increasing electromagnetic field intensities of various plasmonic nanoparticles. Noted that the $g_{lum}$ value of $R$-1UCM/AuNR@Ag659 was able to reach $6.4 \times 10^{-2}$, which was about 43-fold larger than that of $R$-1M (Supplementary Fig. 12). Those results further highlighted the essential role of an electromagnetic field of plasmonic nanoparticles in modulating the intensity of circular polarization.

It has been proven that the enhancement of chiroptical properties originated from the coupling effect between chiral

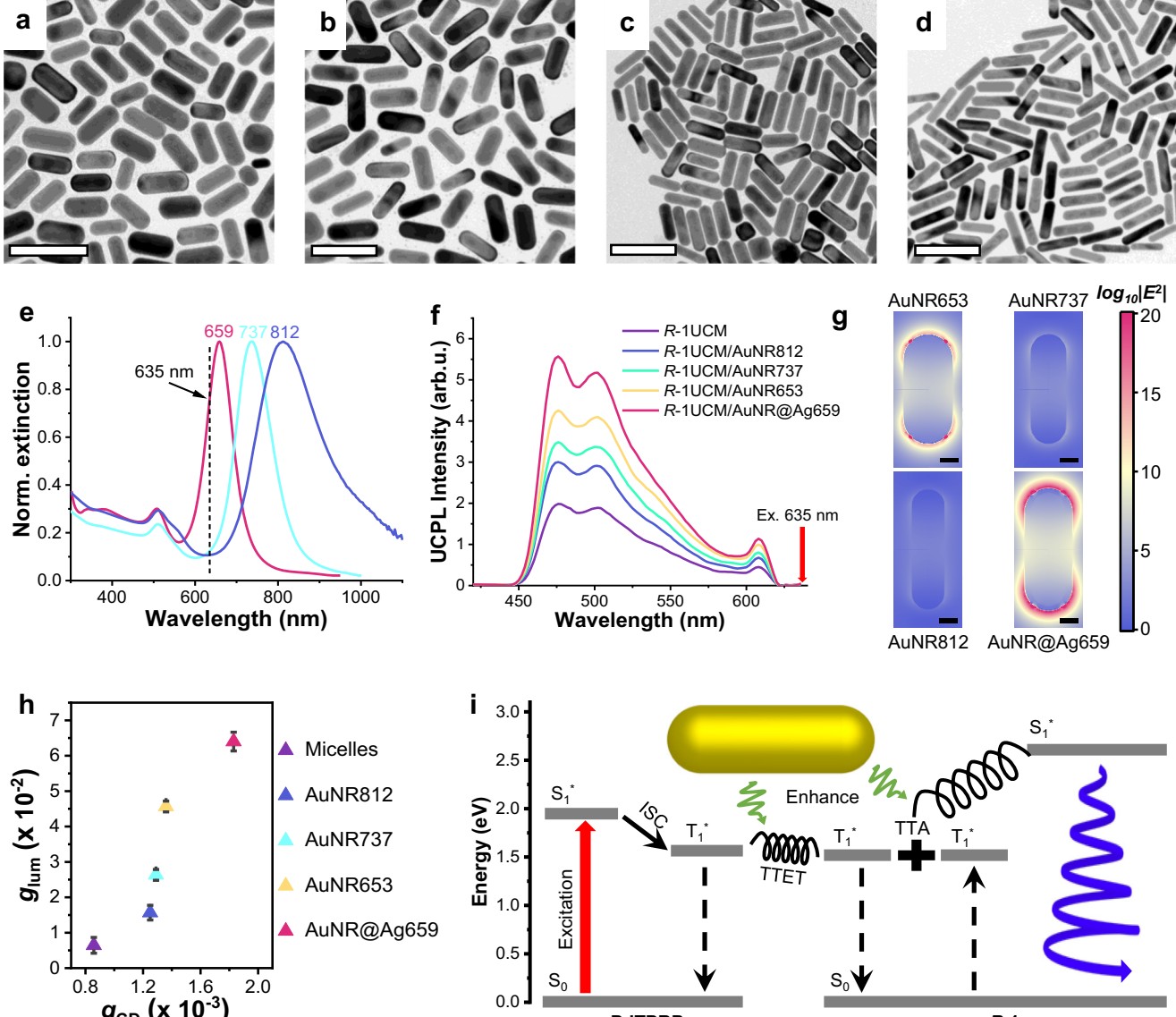

**Fig. 4 | Electromagnetic field-related enhancement of photo upconversion and CPL. a–d** TEM images of **a** AuNR653, **b** AuNR@Ag659, **c** AuNR737, and **d** AuNR812. Scale bar = 200 nm. **e** Normalized extinction spectra of AuNR737 (cyan), AuNR812 (blue), and AuNR@Ag659 (red). The black dashed line indicates the excitation at 635 nm. **f** Upconverted photoluminescence spectra of *R*−1UCM and *R*-1UCM couples with different plasmonic nanorods in deaerated water under excitation of 635 nm laser. [*R*-1] = 5 × 10⁻⁵ mol L⁻¹, [PdTPBP] = 10⁻⁵ mol L⁻¹, [CTAB] = 10⁻² mol L⁻¹, molar ratio AuNR653/*R*−1 = 4.2 × 10⁻⁶/1, 4.2 × 10⁻⁶/1, 3.0 × 10⁻⁶/1, and 2.6 × 10⁻⁶/1 for AuNR812, AuNR737, AuNR653, and AuNR@Ag659, respectively. A 635 nm short-pass filter was set in front of the detector to remove the scattering incident light. **g** Calculated electric field enhancement contours at 635 nm for different nanorods. Scale bar = 10 nm. **h** Plots of $g_{lum}$ versus $g_{CD}$. All error bars show mean ± standard deviation. $n$ = 3 independent experiments. **i** Energy-level diagram for the plasmon-assisted UC-CPL process of *R*−1/PdTPBP fluorophore pair. The black spiral lines represent the chirality-induced spin polarization in the process of TTET and TTA. Source data are provided as a Source Data file.

substances and plasmonic localized electromagnetic field[52,59,60]. However, in a UC-CPL system, one should take into account multiple photoinduced processes. In addition to the well-known promotion of absorption and emission rates, we noted the improved excited-state absorption of *R*-1 in the micelles/plasmon hybrids.

**Table 1 | Size statistics of different plasmonic nanoparticles**

|          | Length/nm   | Width/nm    |
|----------|-------------|-------------|
| AuNR653  | 63.1 ± 14.7 | 28.7 ± 7.5  |
| AuNR737  | 63.2 ± 6.9  | 20.5 ± 3.5  |
| AuNR812  | 69.1 ± 10.8 | 19.1 ± 4.9  |
| AuNR@Ag659 | 71.9 ± 6.9 | 29.1 ± 4.4  |

There were two possible sources of this phenomenon: (1) the enhanced absorption of sensitizers led to an increased generation rate of *R*-1 triplet excitons via TTET and (2) the improved absorption rate of triplet excited state induced by the LSPR effect. Consequently, the total amount of triplet excitons was raised up. Among the chiral upconversion system, the quantitative balance between two spin orientations of triplet excitons was disturbed according to our previous report[25], one kind of triplet excitons with a certain spin orientation was more than its counterpart due to the chirality-induced electron spin polarization in TTET and TTA processes. This type of triplet excitons can be regarded as spin-polarized triplet excitons. In terms of AuNRs assisted UC-CPL process, the amount and fraction of spin-polarized excitons should increase with the increase of the total amount of triplet

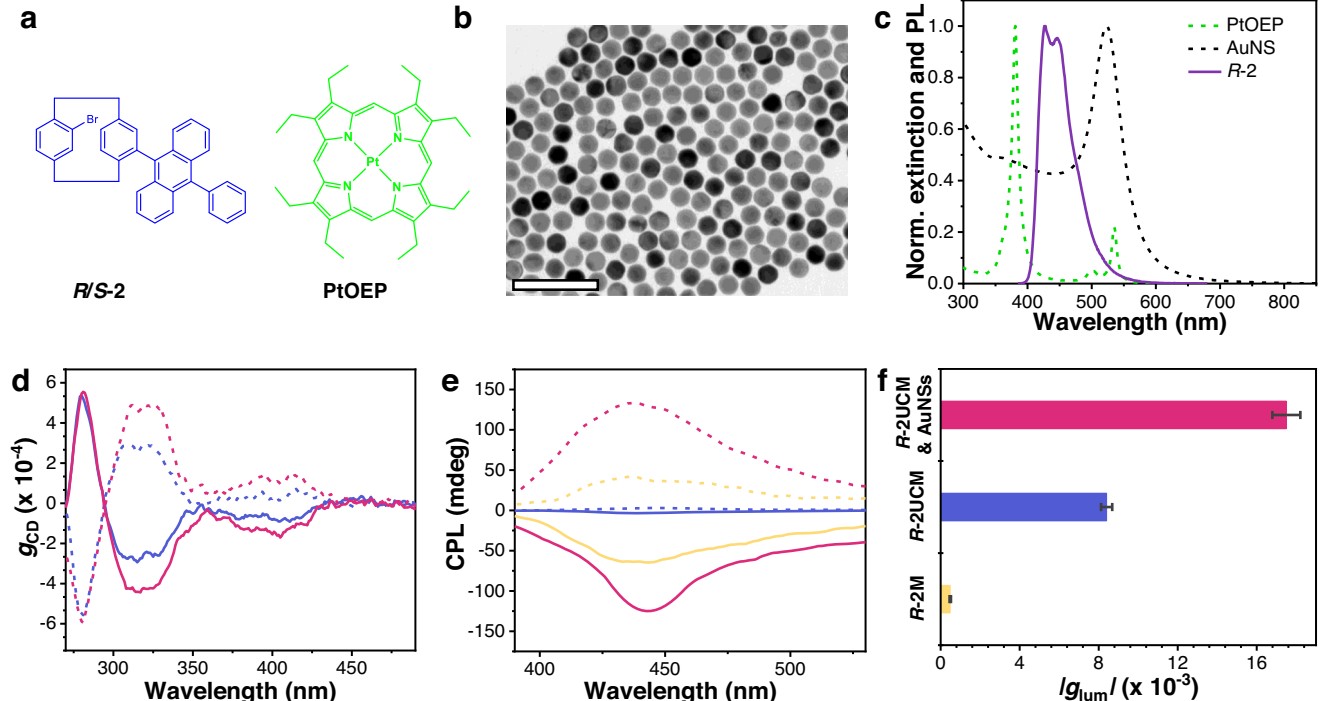

**Fig. 5 | Extension of plasmonic field enhancement effect in chiral emitter 2 systems. a** Molecular structures of chiral emitters R/S-2 and sensitizer PtOEP. **b** TEM image of AuNS. Scale bar = 100 nm. **c** Normalized extinction (dash line) and photoluminescence (solid line) spectra of PtOEP and R-2 in THF and AuNSs in water. $\lambda_{ex}$ = 360 nm. [R-2] = [PtOEP] = $10^{-5}$ mol L$^{-1}$, [AuNSs] = $6.9 \times 10^{-10}$ mol L$^{-1}$. **d** CD dissymmetry factor $g_{CD}$ of R-2M (blue line), S-2M (blue dash line), R-2M/AuNSs (red line), and S-2M/AuNSs (red dash line) versus wavelength. **e** CPL spectra of R-

2M (blue line) and S-2M (blue dash line) in water excited by 360 nm. UC-CPL spectra of R-2UCM (yellow line), S-2UCM (yellow dash line), R-2UCM/AuNSs (red line), and S-2UCM/AuNSs (red dash line) in deaerated water excited by 532 nm laser. [R-2] = $1.3 \times 10^{-4}$ mol L$^{-1}$, [PtOEP] = $10^{-5}$ mol L$^{-1}$, [CTAB] = $10^{-2}$ mol L$^{-1}$, molar ratio AuNR653/R−1 = $3 \times 10^{-6}$/1. **f** Dissymmetry factor $g_{lum}$ of CPL, UC-CPL, and plasmon-enhanced UC-CPL. All error bars show mean ± standard deviation. n = 3 independent experiments. Source data are provided as a Source Data file.

excitons. To verify our hypothesize, we evaluated the intensity of spin polarization by comparing TTA efficiencies between R-1UCM and R-1UCM/AuNR653, because the TTA efficiency ($\Phi_{TTA}$) is significantly affected by the relative density of triplet excitons with opposite spin orientation. Unfortunately, the calculated $\Phi_{TTA}$ of both R-1UCM and R-1UCM/AuNR653 were approximately unit based on the upconversion decay shown in Fig. 3f, because the density of triplet excitons was saturated under high-power excitation. Therefore, we turned to estimate $\Phi_{TTA}$ upon low-power excitation. Interestingly, the $\Phi_{TTA}$ of R-1UCM/AuNR653 (58 %) was lower than that of R-1UCM (67 %, Supplementary Fig. 13), suggesting that the density of triplet excitons with opposite spin orientation was more unbalanced in plasmon-assisted systems. This result agreed with our hypothesis that R-1UCM/AuNR653 probably had a higher spin polarization level. On the one hand, LSPR can enhance the chiral absorption, which was based on electron transition, thus it was also probably to boost the electron spin polarization generated by the energy transfer based on the electron exchange mechanism. Additionally, spin split and magnet-controlled/induced CPL have been reported in pure organic systems[61,62]. These phenomena suggested a possibility for electron spin polarization to have an impact on chiral emission. Therefore, we propose that the effect of LSPR-enhanced chirality-induced spin polarization is critical to amplifying the $g_{lum}$ values of UC-CPL. As shown in Fig. 4i, the chirality-induced spin polarization both in TTET and TTA has been enhanced due to the coupling with plasmon nanorods. Consequently, the UC-CPL resulting from spin-polarized singlet excitons showed a higher $g_{lum}$ value. Indeed, the mechanism proposed still needs a more careful explanation, we suppose this probably will be proved by means of circularly polarized ultrafast spectroscopy in the future.

## Photon upconversion and chiroptical properties enhanced by gold nanospheres

The UC-CPL activity enhanced by LSPR was further verified in another chiral upconversion pairs. We synthesized chiral monomer analogs R-2 and S-2 that were different from dimers R-1 and S-1 (Fig. 5a). In addition, the chromophore changed from a perylene unit to phenyl anthracene. Then micellar aggregates of R-2 (R-2M) were fabricated into assemblies, and the concentration of CTAB in water was kept at $10^{-2}$ mol L$^{-1}$. The maximal fluorescence intensity of R-2M was achieved at an R-2 concentration of $1.3 \times 10^{-4}$ mol L$^{-1}$ (Supplementary Fig. S14). The following experiments were therefore performed at this optimized concentration. Firstly, we investigated the chiroptical responses of R/S-2 in THF and R/S-2M in an aqueous solution. The CD spectra of R-2 exhibited a negative signal around 404 nm, while S-2 showed the opposite curve. There are no obvious changes in the corresponding CD spectra of R/S-2M as well as in CPL spectra (Supplementary Fig. 15), verifying the optical activities of R/S-2M.

Regarding the photon upconversion system, PtOEP was employed as a triplet donor because of the energy-level matching with anthracene derivatives[63,64]. Consequently, gold nanospheres (AuNSs) were applied (Fig. 5b) to couple with the upconverted micelles of R-2 (R-2UCM). As shown in Fig. 5c, there were large overlaps between the plasmon bands of AuNSs and the excitation-emission range of R-2. The upconverted luminescence spectra of R-2UCM showed a blue emission upon excitation of 532 nm laser (Supplementary Fig. 16). After mixing with AuNSs, the upconverted emission of R-2UCM/AuNSs composites showed a 2.7-fold enhancement, which indicated that the LSPR of AuNSs can boost the photon upconversion process. In terms of chiroptical properties, similar to the plasmon complexes of R/S-1M/AuNR653, the ground-state chirality of R/S-2MP/AuNSs was also amplified (Fig. 5d and Supplementary Fig. 17). The |$g_{CD}$| value around 404 nm was amplified from $9.1 \times 10^{-5}$ in R-2M to $1.6 \times 10^{-4}$ in R-2M/

AuNSs. As expected, the TTA-based upconversion could amplify the circular polarization of prompt CPL, then the local electromagnetic fields of metal nanoparticles could further enhance the $g_{lum}$ value of UC-CPL (Fig. 5e). Totally, the $g_{lum}$ value of $R$-2UCMP/AuNSs became 36-time larger than that of prompt CPL of $R$-2M (Fig. 5f). Such behavior suggested the general phenomenon that plasmon could modulate the photon upconversion process and CPL activity.

## Discussion

In summary, we reported an effective and general approach for concurrent boosting upconverted luminescence intensity and $g_{lum}$ value by constructing a composite system composed of chiral upconverted micelles and achiral plasmonic nanoparticles. The generality of plasmon-enhanced UC-CPL was demonstrated by using two types of chiral emitters to couple with different metal nanoparticles. In Au@Ag nanorods and UCM complex system, the overall enhancement of $g_{lum}$ value and upconversion emission intensity has been achieved at 43-time and 3-time, respectively. Based on the theoretical simulations and experimental results, we proposed that plasmon-induced enhancement of absorption of sensitizers, a radiative transition of chiral emitters, and chirality-induced spin polarization were responsible for the plasmon-enhanced optical phenomena. Such plasmon-assisted chiral emitters with high upconversion efficiency and large circular polarization will contribute to chiroptical applications.

## Methods

### Materials

All reagents and solvents were used as received or otherwise indicated. Milli-Q water (18.2 MΩ cm) was used in all the experiments. The chiral emitters $R$-1 and $S$-1 were used in our previous report[25]. Pd(II) meso-tetraphenyl tetrabenzoporphine (>95%) and Pt(II) octaethylporphine (>95%) were purchased from Frontier Scientific, Inc. $R$-4,12-dibromo[2.2]paracyclophane (≥98%, 99% e.e.) and $S$-4,12-dibromo[2.2] paracyclophane (≥98%, 99% e.e.) were purchased from Daicel Chiral Technologies (China) Co., Ltd. Tetrakis(triphenylphosphine)palladium(0) (>97%) and 10-phenyl-9-anthraceneboronic acid (98%) were purchased from TCI (Shanghai) Development Co., Ltd. Poly (styrene sulfonic acid) sodium salt (M.W. 7000 g mol$^{-1}$) was purchased from Alfa Aesar. Chloroauric acid (HAuCl$_4$ • 3H$_2$O, 48–50% Au basis) and silver nitrate (AgNO$_3$, >99%) were purchased from Beijing Chemical Reagent Company. 4-aminophenol (4-ATP, 97%), sodium borohydride (NaBH$_4$, ≥98%), cetyltrimethylammonium bromide (≥98%), L-ascorbic acid (AA, >99%) were purchased from Sigma.

### Characterizations

The $^1$H and $^{13}$C NMR spectra were recorded with a Bruker Fourier 400 (400 MHz) spectrometer, where CDCl$_3$ was used as a solvent and tetramethylsilane was the standard for which $\delta = 0.00$ ppm. UV-vis spectra were obtained using a Hitachi U-3900 spectrophotometer and fluorescence spectra were measured on an Edinburgh FS5 fluorescence spectrophotometer using a Xe lamp as the excitation source. Upconverted luminescence spectra were recorded on a Zolix Omin-λ500i monochromator with a photomultiplier tube (PMTH-R 928) using external excitation source 532 nm (MGL-FN-532, 1.5 W) and 635 nm (MRL-III-635L, 200 mW) lasers (Changchun New Industry Optoelectronic Technology Co. LTD). Upconverted emission decays were recorded on a HORIBA Scientific Nanolog FL3-iHR320 spectrofluorometer using multichannel scaling. CD spectra were measured on a JASCO J-1500 spectrophotometer. CPL spectra were obtained using a JASCO CPL-200 spectrophotometer using a Xe lamp or 375 nm (MDL-III-375, 100 mW, Changchun New Industry Optoelectronic Technology Co. LTD) laser as the excitation source. Upconverted CPL were recorded on a JASCO CPL-200 spectrophotometer with external excitation source 532 nm and 635 nm lasers. Zeta potential and dynamic light scattering were measured on a Zetasizer Nano ZS analyzer. TEM

images were taken using an FEI Tecnai G2 20 S-TWIN microscope (200 kV), and the samples were dropped on carbon-coated Cu grids. Transient absorption spectra were measured on Vitara T-Legend Elite-TOPAS-Helios-EOS-Omni spectrometer supported by Technical Institute of Physics and Chemistry, CAS.

**Theoretical simulation.** The extinction spectra and electric field enhancements of single AuNRs with different sizes and Au@Ag core-shell nanostructure was simulated using FDTD Solutions 8.6 developed by Lumerical Solutions, Inc. The geometric models of individual nanorods are set according to the statistical results that has been summarized in Supplementary Table 2. For the Au@Ag core-shell nanostructure, the thickness of the Ag shell is 0.25 nm for longitudinal and 1.92 nm for transverse sections. An electromagnetic pulse with a wavelength range between 400 and 1000 nm was launched into a box containing the target nanorods/nanostructure to simulate a propagating plane wave interacting with the nanorods/nanostructure. The nanorods/nanostructure and its surrounding medium inside the box were divided into 1 nm meshes. Calculations were done for single nanorod/nanostructure in water. (refractive index of 1.33) and excited by linearly polarized light. The morphology of the nanorods/nanostructure was modeled as a cylinder capped with two half spheres at both ends. For the longitudinal excitation, the incident light was perpendicular to the long axis and polarized along the length axis. The optical constants for bulk Au and Ag were extracted from Johnson Christy database and Palik, respectively.

**Synthesis of $R$/$S$-2.** Synthesis of $R$-1 was performed according to the literature[65]. To a toluene solution (30 mL) containing $R$-4,12-dibromo[2.2]paracyclophane (0.20 g, 0.55 mmol), 10-phenyl-9-anthraceneboronic acid (0.081 g, 0.27 mmol), and tetrakis(triphenylphosphine)palladium(0) (0.032 g, 0.027 mmol), 10 mL Na$_2$CO$_3$ (0.1 M) aqueous solution was added. After degassing the mixture three times by freezing and thawing cycles, the mixed solution was refluxed for 12 h under stirring. After cooling to room temperature, the mixture was concentrated with a rotary evaporator and redissolved in dichloromethane (DCM). Then water was added and the aqueous was extracted with DCM three times. The organic layers were combined, dried over sodium sulfate, and precipitates were removed by filtration. The products were purified by silica gel column chromatography ($n$-hexane/ethyl acetate from 20/1 to 2/1 v/v as an eluent). After evaporating in vacuo, a light-yellow powder of $R$-2 (0.187 g, 0.262 mmol, yield 48%) was collected. $^1$H NMR (400 MHz, CDCl$_3$, 25 °C) δ 9.19 (d, $J$ = 8.8 Hz, 1 H), 7.78 (t, $J$ = 10.0 Hz, 1 H), 7.74 (d, $J$ = 9.6 Hz, 1 H), 7.59 (t, $J$ = 20.8 Hz, 3 H), 7.55 (d, $J$ = 7.2 Hz, 1 H), 7.52 (d, $J$ = 7.2 Hz, 1 H), 7.45 (q, $J$ = 20.0 Hz, 3 H), 7.29 (s, 1 H), 7.22 (t, $J$ = 16.4 Hz, 1 H), 7.14 (t, $J$ = 16.4 Hz, 1 H), 6.93 (d, $J$ = 7.6 Hz, 1 H), 6.89 (s, 1 H), 6.85 (d, $J$ = 4.8 Hz, 1 H), 6.83 (d, $J$ = 4.4 Hz, 1 H), 6.69 (d, $J$ = 7.6 Hz, 1 H), 3.72 (q, $J$ = 22.4 Hz, 1 H), 3.38 (q, $J$ = 23.6 Hz, 1 H), 3.24 (dt, $J$ = 30.8 Hz, 1 H), 3.00 (dt, $J$ = 31.6 Hz, 1 H), 2.90-2.77 (m, 2 H), 2.53-2.39 (m, 2 H). $^{13}$C NMR (101 MHz, CDCl$_3$, 25 °C) δ 141.79, 141.40, 139.71, 139.30, 137.89, 137.31, 135.65, 135.19, 135.17, 134.64, 134.36, 133.49, 131.77, 131.41, 131.24, 131.16, 130.11, 129.21, 129.00, 128.39, 127.39, 127.31, 126.84, 126.63, 125.31, 125.01, 124.83, 124.70, 124.55, 36.71, 34.45, 33.91, 32.70. HR-MS (MALDI) calcd. for C$_{36}$H$_{27}$Br [M$^+$]: 538.129064, found: 538.128934.

Synthesis of $S$-2 was the same as $R$-2, except for $S$-4,12-dibromo[2.2]paracyclophane was used as a chiral reagent. $^1$H NMR (400 MHz, CDCl$_3$, 25 °C) δ 9.19 (d, $J$ = 8.8 Hz, 1 H), 7.78 (t, $J$ = 15.2 Hz, 1 H), 7.74 (d, $J$ = 8.8 Hz, 1 H), 7.60 (t, $J$ = 19.2 Hz, 3 H), 7.56 (d, $J$ = 7.6 Hz, 1 H), 7.52 (d, $J$ = 7.2 Hz, 1 H), 7.46 (q, $J$ = 22.0 Hz, 3 H), 7.29 (s, 1 H), 7.23 (t, $J$ = 16.4 Hz, 1 H), 7.15 (t, $J$ = 16.4 Hz, 1 H), 6.94 (d, $J$ = 7.6 Hz, 1 H), 6.88 (s, 1 H), 6.85 (d, $J$ = 5.2 Hz, 1 H), 6.83 (d, $J$ = 5.2 Hz, 1 H), 6.70 (d, $J$ = 7.2 Hz, 1 H), 3.73 (q, $J$ = 22.4 Hz, 1 H), 3.39 (q, $J$ = 23.6 Hz, 1 H), 3.24 (dt, $J$ = 30.8 Hz,

1 H), 3.01 (dt, $J$ = 31.6 Hz, 1 H), 2.91-2.80 (m, 2 H), 2.53-2.39 (m, 2 H). $^{13}$C NMR (101 MHz, CDCl$_3$, 25 °C) δ 141.79, 141.40, 139.70, 139.30, 137.89, 137.30, 135.65, 135.19, 135.17, 134.64, 134.36, 133.49, 131.76, 131.41, 131.24, 131.16, 130.11, 129.21, 129.00, 128.38, 127.39, 127.31, 126.84, 126.62, 125.31, 125.01, 124.83, 124.70, 124.55, 36.71, 34.44, 33.91, 32.69. HR-MS (MALDI) calcd. for C$_{36}$H$_{27}$Br [M$^+$]: 538.129064, found: 538.128882.

**Synthesis of Au nanoparticles through a seed-mediated growth**
**Preparation of Au seeds.** CTAB-capped seeds were prepared by chemical reduction of HAuCl$_4$ with NaBH$_4$ in the presence of CTAB: a mixture solution was obtained by mixing CTAB (0.1 M, 7.5 mL), HAuCl$_4$ (22.1 mM, 0.113 mL) and deionized water (1.8 mL) under magnetic stirring. Then, a freshly prepared and ice-cold NaBH$_4$ (0.01 M, 0.6 mL) was injected into the mixture solution and continued to agitate for 3 min. The color of the solution turned from yellow to brown after adding NaBH$_4$. The Au-seeds solution was kept at 30 °C without any disturbance for 2–5 h before the further experiment.

**Preparation of gold nanospheres with the diameter around 20 nm.**
(1) About 10 nm Au nanoparticles were synthesized by a seed-mediated method. The growth solution consisted of CTAC (0.1 M, 500 mL), HAuCl$_4$ (22.1 mM, 5.65 mL), and AA (0.1 M, 75 mL). Then, the Au-seeds solution (6.7 mL) was added to the above growth solution to initiate the growth of the gold nanospheres. Then, the well-mixed solution was kept at 30 °C for 0.5 h. The gold nanospheres were purified by centrifugation (24,878×g, 23 min). The precipitates were redispersed in deionized water (50 mL). (2) Synthesis of 20 nm AuNSs. The growth solution consisted of CTAB (1 mM, 500 mL), HAuCl$_4$ (22.1 mM, 5.65 mL), and AA (0.1 M, 20 mL). Then, 12 mL of 10 nm AuNSs was added to the growth solution under magnetic stirring. About 20 nm AuNSs were obtained after overgrowth for 30 min at 40 °C.

**Preparation of AuNR653.** The growth solution of AuNR653 was prepared by mixing CTAB (0.1 M, 100 mL), HAuCl$_4$ (22.1 mM, 2.61 mL), AgNO$_3$ (0.01 M, 0.6 mL), and AA (0.1 M, 0.55 mL) together. Then, Au-seeds solution (120 μL) was added to initiate rod growth at 30 °C. After growth for 12 h, AA (0.1 M, 55 μL) was added twice with a 30 min interval to obtain AuNR653.

**Preparation of AuNR737..** By changing the growth solution to the mixing solution of CTAB (0.1 M, 100 mL), HAuCl$_4$ (22.1 mM, 2.26 mL), AgNO$_3$ (0.01 M, 1.2 mL), and AA (0.1 M, 0.55 mL). The AuNR737 was obtained by using a similar growth procedure to the AuNR653.

**Preparation of AuNR812..** The growth solution consisted of CTAB (0.1 M, 100 mL), HAuCl$_4$ (22.1 mM, 2.26 mL), AgNO$_3$ (0.01 M, 1.0 mL), H$_2$SO$_4$ (0.1 M, 2 mL), and AA (0.1 M, 0.8 mL). Then, the Au-seeds solution (240 μL) was added to the above growth solution to initiate the rod growth. After 12 h, the AuNR812 was obtained.

**Preparation of AuNR@Ag659..** 4-ATP (10 mM, 5 μL) was added in 10 mL AuNR720 (0.1 nM) dispersed in 10 mM CTAB and incubated at a 30 °C water bath for 30 min. Then, AgNO$_3$ (0.1 M, 5 μL) and AA (0.1 M, 50 μL) were added into the above solution to trigger the overgrowth of the Ag shell at 70 °C. After 1 h, AuNR@Ag with the longitudinal SPR band at 659 nm were obtained. All nanorods were purified twice by water with centrifugation (13994 × g, 15 min).

**Preparation of PSS-modified nanoparticles.** The concentrated nanoparticle suspension was mixed with CTAB (0.1 M, 0.01 mL) and incubated at 30 °C for 30 min. Then, a 50 μL mixture solution containing PSS (20 mg/mL) and NaCl (60 mM) was added into the above rod suspension and well-mixed. After incubating at a 30 °C water bath

for 12 h, the PSS-coated nanoparticles were purified by centrifugation twice (13,994×g, 15 min).

**Preparation of micellar aggregates.** All the various micelles were prepared by a typical nanoprecipitation method. For example, 0.06 mL tetrahydrofuran (THF) solution of R-1 (5 mM) was mixed with CTAB (22 mg), then 6 mL water was added rapidly under continuous sonication for 10 min. After that, the mixture was purified by a rotary evaporator at 40 °C to remove THF. The resultant R-1M were further filtered through a 0.22-μm filter membrane.

**Preparation of plasmon-micelle nanocomposites.** The aqueous solution of PSS-modified plasmonic nanoparticles (nanorods and nanospheres) was added into the micelle solution (3 mL) and the micelles were attached on the surface of plasmonic nanoparticles to form plasmon-micelle composites through electrostatic attraction after 10 h.

**Estimation of UCQY.** The unknown upconversion quantum yields ($\Phi_{unk}$) of R-1UCMP and R-1UCMP/AuNR653 were measured with respect to a standard reference solution of methylene blue ($\Phi_{ref}$ = 0.03). The luminescence was measured under the excitation power over 1000 mW cm$^{-2}$, so in the excitation regime of high UC efficiency. The relative quantum yield values $\Phi_{unk}$ were obtained according to the following relation:

$$\Phi_{unk} = \Phi_{ref}\left(\frac{E_{ref}}{E_{unk}}\right)\left(\frac{I_{unk}}{I_{ref}}\right)\left(\frac{n_{unk}}{n_{ref}}\right)^2 \quad (1)$$

where $\Phi_{ref}$ is the reference standard yield, $E$ is the fraction of photons absorbed at the excitation wavelength, $I$ is the integrated photoluminescence intensity and $n$ is the medium refractive index.

**Estimation of TTA efficiency.** The evolution in time of converted luminescence intensity $I_{UC}$ obey to

$$I_{UC} \propto \left[T_A^2\right]_t = \left[T_A^2\right]_0\left(\frac{1-\Phi_{TTA}}{\exp[k_A t]-\Phi_{TTA}}\right)^2 \quad (2)$$

where $T_A$ is the population density of acceptor triplets, $k_A$ is the acceptor triplet's spontaneous decay rates and $k_A = 1/(2 \times \tau_{UC})$, $\Phi_{TTA}$ is annihilation efficiency. It is confirmed that Equation (2) allows to fit properly the decay traces to give annihilation efficiency[28].

**Reporting summary**
Further information on research design is available in the Nature Portfolio Reporting Summary linked to this article.

## Data availability
The authors declare that all data supporting the findings of this study are available within this article and Supplementary Information files and from the corresponding author upon request. Source data are provided with this paper.

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

## Acknowledgements

This work was supported by the National Key Basic R&D Research Program of the Ministry of Science and Technology of the People's Republic of China (2021YFA1200303, P.D.); the Strategic Priority Research Program of Chinese Academy of Sciences (XDB36000000, X.W. and P.D.); the National Natural Science Foundation of China (22172041, P.D., 52173159, P.D., 91856115, P.D.); the Beijing Natural Science Foundation (JQ21003, P.D.).

## Author contributions

P.D. and X.W. conceived the idea and supervised the project. T.Z., D.M., P.D., and X.W. designed the experiments. T.Z. and D.M. performed the chiroptical study. T.Z., D.M., W.S., Y.J., J.H., and X.J. designed, synthesized, and characterized the materials. Z.H. carried out the theoretical calculations. T.Z., D.M., Z.H., X.W., and P.D. wrote and revised the manuscript. All authors analyzed and discussed the results and have given approval for the final version of the manuscript.

## Competing interests

The authors declare no competing interests.
