## [Peer Review File · Nature Communications]

Reviewer comments, first round

Reviewer #1 (Remarks to the Author):

In this manuscript, the authors demonstrated that the upconverted emission and CPL activity in the TTA-UC systems consisting of sensitizers and chiral emitters are enhanced by the LSP resonance of anisotropic metal nanoparticles. The upconverter emission was enhanced up to 6.7 times by hybridizing the nanoscale micellar system consisting of perylene derivative as a chiral emitter and the Pd porphyrin derivative as a sensitizer with Au nanorods whose LSP resonance wavelength mainly overlapped with the photoexcitation wavelength of the sensitizer. And the glum was amplified 7.5 times, as compared with the non-plasmonic reference system (R-1UCMP). The authors claimed that the enhancement factors are mainly due to the enhancement of the photoexcitation efficiency of the sensitized molecules and the LSP resonance-enhanced chirality-induced spin polarization. These were supported by the transient spectral properties of the Au nanorod-UCMP hybrid systems and correlation between and the electromagnetic field intensity of Au nanorods and Au/Ag nanorods and the enhancement of glum and gCD. The mechanism for the enhancement of CPL activity claimed in this manuscript may be interesting, but poorly supported by the experiments and discussion. In addition, some minor but important experimental validations are missing from this study. The reviewer finds it difficult to judge at this stage whether the manuscript is suitable for publication in Nature Communications. The reviewer would like to review again after the authors have made appropriate revisions for the following comments.

1. The authors suggest the rapid accumulation of R-1 triplet excitons (spin-polarized triplet excitons) as a mechanism for the highly activated CPL by the LSP resonance of Au nanorods. The discussion on this is too vague for the reviewer to make a valid judgement on the claim by the authors. For the rapid accumulation of triplet excitons, the reviewer thinks that it is not necessary to develop LSP resonance at the photoexcitation wavelength of the sensitizer. Or does the rapid accumulation of triplet excitons involve the enhancement of the photoexcitation efficiency of the sensitizer? The reviewer believes that the authors should discuss this in detail. The reviewer considers this problem to be a critical point in this manuscript.
2. There is a lack of experimental identification regarding the fabrication of hybrids consisting of Au nanorods and the TTA-UC micelle system. For example, the authors claim that the extinction spectrum of Au nanorods show an effective overlap between their longitudinal mode of LSP resonance and the photoexcitation wavelength of sensitizer. However, the extinction spectrum of hybrids of Au nanorods and TTA-UC micelle systems should be measured because the resonance wavelength in the longitudinal mode of Au nanorods is highly dependent on the refractive index environment of the surface. This is a common problem for all Au and Au/Ag nanorods used in this study.
3. It is difficult to determine whether hybrids of Au nanorods and the TTA-UC micelle system has been formed. The authors have determined the formation of the hybrids only by the zeta potential of the micelle system and TEM image of the hybrids after hybridizing the Au nanorods with the micelle system. This is not sufficient. In particular, the formation of the hybrids cannot be determined from the TEM image of the hybrids. Was there any change in the zeta potential of the Au nanorods after the hybridization. In addition, did the DLS measurements confirm an increase in the hydration radius of the Au nanorods along with the formation of the hybrids.
4. On page 4 in the manuscript, the authors described that the glum value increased to 4.6×10^{-2} (molar ratio of Au nanorods and R-1: $3.0 \times 10^{-6}/1$) with the formation of Au nanorods and the TTA-UC micelle system. However, on page 6, the authors described that the glum value of the same complex increased to 4.8×10^{-2} (molar ratio of AuNRs and R-1: $3.0 \times 10^{-9}/1$). The latter appears to be the optimal complex condition, but I do not understand the intent of the statement on page 4.
5. As shown in Figure 2, the hybridization of TTA-UC micelle system and Au nanorods enhanced

the UC emission by a factor of 6.7, while the threshold excitation intensity decreased from 1637 mWcm⁻² to 1318 mWcm⁻². What effect of LSP resonance do the authors believe is responsible for the decrease in threshold excitation light intensity? The reviewer is concerned about the discrepancy between the large enhancement of UC emission and the slight decrease in the threshold excitation power.

6. The authors claim that one mechanism for the enhancement of UC emission is the amplification of the radiative rate of chiral emitter. However, the reviewer believe that the quantum yields of the chiral emitter synthesized by the authors will need to be measured in order to verify the enhancement by the phenomenon.

7. The authors have synthesized Au/Ag nanorods from the Au653 as a core, but their LSP resonance wavelength are only slightly changed. Normally, the LSP resonance wavelength of Au nanorods is shifted to a shorter wavelength by the coating of Ag thin film. Was the thickness of the Ag thin film of the synthesized Au/Ag nanorods measured? The reviewer cannot know what geometric model was used in the FDTD calculation.

Reviewer #2 (Remarks to the Author):

In this manuscript, the authors focus on plasmonic nanoparticles-integrated chiral nanocomposites for the enhancement of upconverted photoluminescence and circularly polarized luminescence (CPL). Localized surface plasmon resonance (LSPR) of plasmonic nanoparticles affects three processes; the absorption of sensitizers, the triplet-triplet energy transfer, and the radiative decay of emitters. Both upconverted emission and CPL are amplified due to the LSPR of plasmonic nanoparticles. The authors demonstrate that the luminescence dissymmetry factor is enhanced 43 times by combining the chiral emitters with the core-shell nanoparticles supporting LSPR near 635 nm. The method using LSPR in triplet-triplet annihilation-based photon upconversion systems sounds interesting. This manuscript is also well-organized and clear. Nevertheless, it has several concerns that should be addressed.

1. The authors report that the core-shell plasmonic nanoparticle has a stronger response. Is there any relationship between the core-shell structure and LSPR? LSPR is the resonance mode related to the surface plasmons so I wonder why the core-shell structure can make the surface plasmon resonance more intense. I suggest that authors add the detailed explanation in this manuscript.

2. The authors make the dominant resonance mode of plasmonic nanoparticles overlap with the wavelength of the excitation laser. I am just curious why the author did not design LSPR for the emission to be dominant. Even when the resonant mode is targeted to the emission wavelength, the photoluminescence can be enhanced and the lifetime is reduced as in Figure 3d. Rather, more dramatic enhancement can be obtained. Would you explain reasons for the design of plasmonic nanoparticles?

3. Several optical transition processes are considered to prove the applicability of the method using LSPR. The authors need to provide individual energy-level diagrams of sensitizers and chiral emitters, which can explain types of transitions. Although comprehensive photophysical processes of TTA-UC systems are explained in Figure 1, additional figures of intuitive energy-level diagrams help readers to understand upconversion emission and CPL spectra easily.

4. The authors may add the recent papers about chirality, especially, theoretical studies, fundamentals and optics, such as

<https://www.nature.com/articles/s41377-020-00367-8>

<https://pubs.acs.org/doi/full/10.1021/acsp Photonics.1c00882>

<https://pubs.rsc.org/en/content/articlelanding/2021/nr/d1nr05805c>

Overall, I think this manuscript can be published in Nature Communications after these concerns being addressed.

Reviewer #3 (Remarks to the Author):

This work is about enhancement of circularly polarized luminescence (CPL) in a complex molecular-plasmonic nano-system. The authors explored a mixed sensitizer-emitter luminescence upconversion micelle system, which on coating gold nanorods, produces enhanced luminescence and CPL. The enhanced luminescence in such systems was reported before, and the novelty in this work is the introduction of chirality in the molecules and observation of enhanced CPL. I would say that there is interest in finding ways to enhance CPL by plasmonic systems, as reported in this work. However, I feel that the publication of this work in Nature Communications would require more information and better mechanistic discussion, as detailed here:

1. There is not enough CPL data presented to understand what is going on in the studied system. The enhanced CPL spectra shown in Fig. 3A are fairly noisy for a supposedly strong CPL and much broader than the emission lines themselves, which makes one a bit suspicious about the measured CPL source, and whether there might be experimental artifacts in the measurements. In addition no glum spectra are shown, only peak values plotted in Fig. 4d. I suspect that the glum spectra would be also very noisy and they should be added, at least in the Supplementary Information, to understand if they have a physical meaning. Including the glum spectrum for the Au@Ag nanorod system, which is supposed to be the record one.
2. The explanation about connection of spin-polarization in the energy transfer process and the enhanced CPL is not clear to me. Eventually, the CPL magnitude (and glum) is determined by the properties of the emitting state alone. The above-mentioned explanation of the authors is valid only if the emitting state is spin-split and its sub-levels are selectively populated depending on the spin-state of the transferred electrons. Is this the case? Is there evidence for this? The CPL spectra are too noisy to decide whether there are spectral shifts between the two enantiomers, which might be the required evidence for the splitting in the emitting levels.
3. Is it possible that changes in CPL on attaching the micelles to the nanorods occur due to reorganization of the molecules in the micelles? I think that the authors should discuss this reasonable possibility and try to bring evidence for/against it.
4. The dynamics of the excited states is less interesting here. The fact that lifetimes are shortened by the presence of metal nanostructures is well known and I think that it does not contribute much to the understanding of CPL effects. I would reduce this part and improve discussion of the above three points.

After addressing the above issues the work could be considered again for publication.

Reviewer #4 (Remarks to the Author):

The manuscript by Duan et al. reports an upconverting system showing CPL. η_{glum} factors are significantly enhanced upon adsorption to metal surface, such as Au rods. The main and most novel point of the work is that plasmon resonance can amplify CPL associated to TTA-UC (triplet-triplet annihilation-upconversion). Given the interest of a large community on these topics, I think that the manuscript could be accepted after careful revisions.

- 1) It is not entirely clear to me how plasmon resonance works to amplify TTA-UC CPL. What is the effect of plasmon resonance on regular (down-converted) CPL? It would be useful to see a full comparison between regular and UC CPL for the complete assembly (system/Au). Moreover, Why the presence of Au nanoparticles has only little effects on the CD properties? This is an important point, since the authors claim that "either ground- or excited-state chirality can be enhanced by plasmonic LSPR effect".
- 2) "Chirality-induced spin polarization", please clarify.
- 3) "The upconverted luminescence intensity versus the concentration of metal nanoparticles showed a volcano shape" please clarify.
- 4) In Fig. 3 and elsewhere, excitation wavelengths should be reported.

5) The free micelle concentration should be estimated to have an accurate value for g factors/enhancement.

6) "Due to the competed absorption", it should read "competing".

7) "Indicating that the formation of micelle remained the chirality of chiral emitters", please check this sentence.

8) The manuscript should be checked throughout for grammar mistakes and to improve clarity in general.

Reviewer #1 (Remarks to the Author):

In this manuscript, the authors demonstrated that the upconverted emission and CPL activity in the TTA-UC systems consisting of sensitizers and chiral emitters are enhanced by the LSP resonance of anisotropic metal nanoparticles. The upconverter emission was enhanced up to 6.7 times by hybridizing the nanoscale micellar system consisting of perylene derivative as a chiral emitter and the Pd porphyrin derivative as a sensitizer with Au nanorods whose LSP resonance wavelength mainly overlapped with the photoexcitation wavelength of the sensitizer. And the glum was amplified 7.5 times, as compared with the non-plasmonic reference system (R-1UCMP). The authors claimed that the enhancement factors are mainly due to the enhancement of the photoexcitation efficiency of the sensitized molecules and the LSP resonance-enhanced chirality-induced spin polarization. These were supported by the transient spectral properties of the Au nanorod-UCMP hybrid systems and correlation between and the electromagnetic field intensity of Au nanorods and Au/Ag nanorods and the enhancement of glum and gCD.

The mechanism for the enhancement of CPL activity claimed in this manuscript may be interesting, but poorly supported by the experiments and discussion. In addition, some minor but important experimental validations are missing from this study. The reviewer finds it difficult to judge at this stage whether the manuscript is suitable for publication in Nature Communications. The reviewer would like to review again after the authors have made appropriate revisions for the following comments.

We thank the reviewer for their time in assessing the manuscript and the insightful comments given. The revisions are detailed below.

1. The authors suggest the rapid accumulation of R-1 triplet excitons (spin-polarized triplet excitons) as a mechanism for the highly activated CPL by the LSP resonance of Au nanorods. The discussion on this is too vague for the reviewer to make a valid judgement on the claim by the authors. For the rapid accumulation of triplet excitons, the reviewer thinks that it is not necessary to develop LSP resonance at the photoexcitation wavelength of the sensitizer. Or does the rapid accumulation of triplet excitons involve the enhancement of the photoexcitation efficiency of the sensitizer? The reviewer believes that the authors should discuss this in detail. The reviewer considers this problem to be a critical point in this manuscript.

Response: We thank the reviewer for this insightful comment. We agree that the enhancement of the photoexcitation efficiency is one of the reasons for the increasing amount of triplet excitons. Additionally, another source for the increasing triplet excitons should be the improved excited-state absorption rate that induced by LSPR effect. In fact, the overlap of LSPR and photoexcitation wavelength is very important, as we found that the R-1UCMP/Au653 showed the best UC-CPL property than R-1UCMP/Au737 and R-1UCMP/Au812. The longitudinal LSPR wavelengths of the latter two systems are less matching with the excitation light, leading to weak local electromagnetic fields, which results in dissatisfactory enhancement effect. We have

tried to give a clearer explanation for the plasmon-enhanced UC-CPL on page 10:

Addition to the well-known promotion of absorption and emission rates, we noted the improved excited-state absorption of *R*-1 in the micelle/plasmon hybrids. There were two possible sources of this phenomenon: (1) the enhanced absorption of sensitizers led to an increasing generation rate of *R*-1 triplet excitons via TTET, (2) the improved absorption rate of triplet excited state induced by LSPR effect. Consequently, the total amount of triplet excitons was raised up. Among the chiral upconversion system, the quantitative balance between two spin orientations of triplet excitons was disturbed according to our previous report,²⁵ one kind of triplet excitons with certain spin orientation was more than its counterpart due to the chirality-induced electron spin polarization in TTET and TTA processes. This type of triplet excitons can be regarded as spin-polarized triplet excitons. In terms of AuNRs assisted UC-CPL process, the amount and percentage of spin-polarized triplet excitons should be increasing with the above-mentioned growth of triplet excitons. On the one hand, LSPR can enhance the chiral absorption which was based on electron transition, thus it was also probably to boost the electron spin polarization generated by the energy transfer that based on the electron exchange mechanism. Additionally, spin split and magnet-controlled/induced CPL have been reported in pure organic systems.^{61,62} These phenomena suggested a possibility for electron spin polarization to have an impact on chiral emission. Therefore, we proposed that the effect of LSPR-enhanced chirality-induced spin polarization played an important role in amplifying the g_{lum} values of UC-CPL. As shown in Fig. 4e, the chirality-induced spin polarization both in TTET and TTA has been enhanced due to the coupling with plasmon nanorods. Consequently, the UC-CPL resulted from spin-polarized singlet excitons showed a higher g_{lum} value.

2. There is a lack of experimental identification regarding the fabrication of hybrids consisting of Au nanorods and the TTA-UC micelle system. For example, the authors claim that the extinction spectrum of Au nanorods show an effective overlap between their longitudinal mode of LSP resonance and the photoexcitation wavelength of sensitizer. However, the extinction spectrum of hybrids of Au nanorods and TTA-UC micelle systems should be measured because the resonance wavelength in the longitudinal mode of Au nanorods is highly dependent on the refractive index environment of the surface. This is a common problem for all Au and Au/Ag nanorods used in this study.

Response: We agree that the certainty that the fabrication of micelles/AuNRs hybrids is important. As show in Fig. R1, the LSPR wavelengths of plasmon nanorods exhibit slight changes, which might be due to the attaching of *R*-1UCMP. However, the overlap between their longitudinal LSPR and the photoexcitation wavelength of sensitizer is still effective. We are confident that the LSPR activities of these plasmonic nanorods are still stable in the hybrid systems. We added the data and corresponding description in the revised manuscript and supplementary information, the sentence on page 9 reads:

Moreover, the LSPR activities of these nanorods remained stable as their extinction spectra only showed slight shift or broadening after hybrid process (Supplementary Fig. 12).

Figure R1. Normalized extinction spectra of various *R*-1UCMP/AuNRs.

3. It is difficult to determine whether hybrids of Au nanorods and the TTA-UC micelle system has been formed. The authors have determined the formation of the hybrids only by the zeta potential of the micelle system and TEM image of the hybrids after hybridizing the Au nanorods with the micelle system. This is not sufficient. In particular, the formation of the hybrids cannot be determined from the TEM image of the hybrids. Was there any change in the zeta potential of the Au nanorods after the hybridization. In addition, did the DLS measurements confirm an increase in the hydration radius of the Au nanorods along with the formation of the hybrids.

Response: As responded to comment 2, we agree that the suggested characterizations are also all useful and have done the following additional experimentation. In Fig. R2a, the potential of the Au653 is negatively charged and the micelle is positively charged. After the hybrid complex is formed, the potential of *R*-1UCMP/Au653 is lower than the micelle solution. In order to exclude the influence of free micelle, the complex precipitate, consisting of Au653 and the attached *R*-1UCMP, is re-dissolved in water after centrifugate. It's clear to see that the negatively charged Au653 convert to +30 mV because of the hybridizing with *R*-1UCMP. As for the DLS data, the hydrodynamic diameters of Au653 grow from 4.9 and 58.8 nm to 7.5 and 68.1 nm in *R*-1UCMP/Au653 system (Fig. R2b), also indicating the successful fabrication of those hybrids. As TEM image is recorded in a solid state, it's actually not enough to prove the formation of the hybrids in solution. Thus, we have deleted this description and added these new results in the revised manuscript.

On page 5:

The mixed solution was kept undisturbed for 10 hours, and the Zeta potential decreased from +67 mV to +58 mV after Au653 added into the *R*-1UCMP solution (Fig. 2c). To further identify the successful formation of *R*-1UCMP/Au653 hybrids, the precipitate was collected after centrifugation at 8000 rpm for 10 min and redispersed with water. The Zeta potential of the composites was +30 mV while a pure Au653 was -12 mV. Moreover, in *R*-1UCMP/Au653 system, a growing of hydrodynamic diameters of Au653 from 4.9 and 58.8 nm to 7.5 and 68.1 nm was studied by dynamic light scattering (Supplementary Fig. 4). Therefore, the electrostatic attraction enabled attaching of the positively charged *R*-1UCMP to the surface of the negatively charged Au653.

Figure R2. a) Zeta potential of Au653 and precipitate of *R*-1UCMP/Au653 dispersed in aqueous solution. b) Particle size distribution for a stable dispersion of Au653 and *R*-1UCMP/Au653 in water obtained by DLS.

4. On page 4 in the manuscript, the authors described that the g_{lum} value increased to 4.6×10^{-2} (molar ratio of Au nanorods and R-1: $3.0 \times 10^{-6}/1$) with the formation of Au nanorods and the TTA-UC micelle system. However, on page 6, the authors described that the g_{lum} value of the same complex increased to 4.8×10^{-2} (molar ratio of AuNRs and R-1: $3.0 \times 10^{-9}/1$). The latter appears to be the optimal complex condition, but I do not understand the intent of the statement on page 4.

Response: We apologize to the confusion caused. On page 4, we present the g_{lum} value of *R*-1UCMP/Au653 with the molar ratio of $mol_{Au653} : mol_{R-1} = 3.0 \times 10^{-6} : 1$, because of the concurrent enhancement of the g_{lum} value and upconversion emission intensity at such condition. This part of the Introduction reads:

When the molar ratio of AuNRs and *R*-1 was $3.0 \times 10^{-6}/1$, the g_{lum} value increased to 4.6×10^{-2} . At the same time, the upconverted emission intensity of *R*-1UCMP/AuNRs was approximately two times larger than that of the pristine upconversion micelles.

On page 6, we are exploring the concentration dependence of g_{lum} values, and when the molar ratio of Au653 to *R*-1 is $3.9 \times 10^{-6} : 1$, the g_{lum} value is exactly a little bit higher

than that one of $\text{mol}_{\text{Au653}} : \text{mol}_{R-1} = 3.0 \times 10^{-6} : 1$. However, the upconverted emission intensity is suppressed in this condition. So we select $\text{mol}_{\text{Au653}} : \text{mol}_{R-1} = 3.0 \times 10^{-6} : 1$ as the research object for the following investigation and discussion. To make this information clear, we revised some description in the manuscript, these parts are:

On page 5,

When the ratio of Au653 and R-1UCMP reached to 3.9×10^{-6} , upconverted luminescence intensity would decrease due to the competing absorption at 635 nm between sensitizer and Au653 and the reabsorption of upconverted luminescence by Au653.

and on page 6,

With the addition of Au653, the g_{lum} values of PUC-CPL gradually increased and reached the maximum when $\text{mol}_{\text{Au653}}/\text{mol}_{R-1}$ is $3.9 \times 10^{-6}/1$,

and on page 7,

In this operation, the R-1UCMP/Au653 composite with a $\text{mol}_{653}/\text{mol}_{R-1}$ at $3.0 \times 10^{-6}/1$ was selected due to high-concentration Au653 can quench the upconverted emission.

and on page 9,

The molar ratio of Au653 to R-1 was fixed at $3.0 \times 10^{-6}/1$.

Therefore, we prefer to take this system (molar ratio of Au653 to R-1 is $3.0 \times 10^{-6} : 1$) as a representation in the Introduction.

5. As shown in Figure 2, the hybridization of TTA-UC micelle system and Au nanorods enhanced the UC emission by a factor of 6.7, while the threshold excitation intensity decreased from 1637 mWcm^{-2} to 1318 mWcm^{-2} . What effect of LSP resonance do the authors believe is responsible for the decrease in threshold excitation light intensity? The reviewer is concerned about the discrepancy between the large enhancement of UC emission and the slight decrease in the threshold excitation power.

Response: The upconverted emission is improved due to the LSPR effect which can promote the absorption of sensitizer and the radiative rate of emission. However, the threshold power intensity (I_{th}) depends on triplet decay rate of acceptor (k^{T}), absorption coefficient of sensitizer (α), triplet energy transfer efficiency (Φ_{ET}), and second-order rate constant of the TTA process (k_{TTA}):

$$I_{\text{th}} = (k^{\text{T}})^2 / \alpha \Phi_{\text{ET}} k_{\text{TTA}}, \text{ in which the triplet decay rate } k^{\text{T}} = 1/\tau^{\text{T}} = 1/(2\tau_{\text{UC}}).$$

In the hybrid systems, we assume that the Φ_{ET} and k_{TTA} have negligible changes because the LSPR is more effective to absorption and emission, the core factors to determine the I_{th} are k^{T} and α . We already know that the α of sensitizer increases due to the LSPR effect, which can lower the I_{th} . However, as shown in Fig. 3d, the lifetime of UC shortens to $82 \mu\text{s}$, thus the k^{T} increases, which can raise up the I_{th} . Therefore, there is an antagonistic effect between k^{T} and α to decrease the I_{th} , which could be the main reason for the slight decrease in the threshold excitation power.

6. The authors claim that one mechanism for the enhancement of UC emission is the

amplification of the radiative rate of chiral emitter. However, the reviewer believe that the quantum yields of the chiral emitter synthesized by the authors will need to be measured in order to verify the enhancement by the phenomenon.

Response: As suggested by reviewer, we have tested the quantum yields both of chiral emitters (35%) and chiral micelles (18%) and added these data in the revised manuscript (highlighted in yellow on page 4) and supplementary information (as Supplementary Table 1). Unfortunately, for the hybrid systems, it is still a challenge to distinguish between the plasmon resonance absorption and the fluorophore absorption in the measurement. So, we apologize that we can't get more evidence from the quantum yields. However, the reduction of UC lifetime is also a good evidence for the promotion of the radiative rate of chiral emitter. Although in principle the radiative and nonradiative decay pathways are both possible to shorten the UC lifetime, the alkyl chain lengths of PSS and CTAB is sufficient to make a proper distance between chiral emitters and AuNRs, which allows the radiative rate enhancement via surface plasmons to become dominant (*Chem. Commun.*, 2014, 50, 11169—11172; *J. Phys. Chem. C* 2014, 118, 6398-6404; *J. Phys. Chem. Lett.* 2012, 3, 191–202). Therefore, the faster radiative rate is one of the reasons for the UC emission enhancement.

7. The authors have synthesized Au/Ag nanorods from the Au653 as a core, but their LSP resonance wavelength are only slightly changed. Normally, the LSP resonance wavelength of Au nanorods is shifted to a shorter wavelength by the coating of Ag thin film. Was the thickness of the Ag thin film of the synthesized Au/Ag nanorods measured? The reviewer cannot know what geometric model was used in the FDTD calculation.

Response: The reviewer is right that the LSPR resonance wavelength of Au nanorods is shifted to a shorter wavelength by the coating of Ag thin film. In this present work, the Au@Ag659 is synthesized from the Au720 as the core instead of the Au653, the detailed method that has been showed in Supplementary Information is:

6. Preparation of Au@Ag nanorods with the longitudinal SPR band around 659 nm

4-ATP (10 mM, 5 μ L) was added in 10 mL Au720 (0.1 nM) dispersed in 10 mM CTAB and incubated at 30 $^{\circ}$ C water bath for 30 min. Then, AgNO₃ (0.1 M, 5 μ L) and AA (0.1 M, 50 μ L) were added into the above solution to trigger the overgrowth of Ag shell at 70 $^{\circ}$ C. After 1 h, Au@AgNRs with the longitudinal SPR band at 659 nm were obtained.

All nanorods were purified by centrifugation twice (12000 rpm, 15 min).

The thickness of the Ag shell is 1.92 nm for longitudinal and 0.25 nm for transverse sections. We have added the geometric model in the method of FDTD simulation:

Theoretical Simulation: The extinction spectra and electric field enhancements of single AuNR with different sizes and Au@Ag core-shell nanostructure was simulated using FDTD Solutions 8.6 developed by Lumerical Solutions, Inc. The geometric models of individual nanorods are set according to the statistical results that has been summarized in Supplementary Table 2. For the Au@Ag core-shell nanostructure, the thickness of Ag shell is 0.25 nm for longitudinal and 1.92 nm for transverse sections. An electromagnetic pulse with a wavelength range between 400 nm and 1000 nm was launched into a box containing the target nanorods/nanostructure to simulate a propagating plane wave interacting with the nanorods/nanostructure. The nanorods/nanostructure and its surrounding medium inside the box were divided into 1 nm meshes. Calculations were done for single nanorod/nanostructure in water (refractive index of 1.33) and excited by linearly polarized light. The morphology of the nanorods/nanostructure was modeled as a cylinder capped with two half spheres at both ends. For the longitudinal excitation, the incident light was perpendicular to the length axis and polarized along the length axis. The optical constants for bulk Au and Ag were extracted from Johnson Christy database and Palik, respectively.

Reviewer #2 (Remarks to the Author):

In this manuscript, the authors focus on plasmonic nanoparticles-integrated chiral nanocomposites for the enhancement of upconverted photoluminescence and circularly polarized luminescence (CPL). Localized surface plasmon resonance (LSPR) of plasmonic nanoparticles affects three processes; the absorption of sensitizers, the triplet-triplet energy transfer, and the radiative decay of emitters. Both upconverted emission and CPL are amplified due to the LSPR of plasmonic nanoparticles. The authors demonstrate that the luminescence dissymmetry factor is enhanced 43 times by combining the chiral emitters with the core-shell nanoparticles supporting LSPR near 635 nm. The method using LSPR in triplet-triplet annihilation-based photon upconversion systems sounds interesting. This manuscript is also well-organized and clear. Nevertheless, it has several concerns that should be addressed.

1. The authors report that the core-shell plasmonic nanoparticle has a stronger response. Is there any relationship between the core-shell structure and LSPR? LSPR is the resonance mode related to the surface plasmons so I wonder why the core-shell structure can make the surface plasmon resonance more intense. I suggest that authors add the detailed explanation in this manuscript.

Response: Thanks for your comments. There is no relationship between the core-shell structure and the LSPR intensity. In our manuscript, the core-shell structures own higher plasmonic response intensity due to the Ag shell, we used core-shell structure to obtain stable Ag nanorods with higher LSPR response. Compared with Au, Ag possess stronger plasmonic responses (*J. Phys. Chem. B* 2001, 105, 33, 7871–7873). Also we have verified in our previous research (*Adv. Optical Mater.* 2021, 9, 2001274). We synthesized AuNR@Ag and AuNR@Au with the same thickness of Ag and Au shell, respectively. But the AuNR@Ag showed stronger intensity (Fig. R3).

The explanation on page 9 reads:

The rest one was gold-core-silver-shell nanorod with longitudinal LSPR band at 659 nm, and this core-shell nanoparticle was expected to have a stronger plasmonic responses due to the existence of silver.^{37,58}

Figure R3. The extinction of AuNR@Au, AuNR@Ag and AuNR.

2. The authors make the dominant resonance mode of plasmonic nanoparticles overlap

with the wavelength of the excitation laser. I am just curious why the author did not design LSPR for the emission to be dominant. Even when the resonant mode is targeted to the emission wavelength, the photoluminescence can be enhanced and the lifetime is reduced as in Figure 3d. Rather, more dramatic enhancement can be obtained. Would you explain reasons for the design of plasmonic nanoparticles?

Response: Thank reviewer for this comment. We have explored different overlap degrees between the absorption (excitation light) and longitudinal LSPR by using Au653, Au737, and Au812 nanorods to couple with *R*-1UCMP. Meanwhile, the transverse LSPR of these nanorods match with the emission of *R*-1UCMP. However, the enhancement of UC emission gradually decreases along with the decreasing overlaps between longitudinal LSPR and excitation light. In addition, although the matching of emission and the transverse LSPR existing, the regular downshifting CPL of *R*-1MP/Au653 that is excited by 400 nm light only shows slight enhancement. Because the electromagnetic field of Au653 is too weak under such excitation case. A related investigation was also described in *Adv. Funct. Mater.* 2017, 27, 1701842. Therefore, we think the dual excitation and emission enhancements are more reasonable.

3. Several optical transition processes are considered to prove the applicability of the method using LSPR. The authors need to provide individual energy-level diagrams of sensitizers and chiral emitters, which can explain types of transitions. Although comprehensive photophysical processes of TTA-UC systems are explained in Figure 1, additional figures of intuitive energy-level diagrams help readers to understand upconversion emission and CPL spectra easily.

Response: We appreciate the reviewer's suggestion. We have drawn the energy-level diagrams of sensitizers and chiral emitters, as well as the transition process involving in the UC-CPL. This figure is shown below (Fig. R4) and added in the revised manuscript as Fig. 4e.

Figure R4. Energy-level diagram for the plasmon-assisted UC-CPL process of *R*-1/PdTPBP fluorophore pair. The black spiral lines represent the chirality-induced spin polarization in the process of TTET and TTA.

4. The authors may add the recent papers about chirality, especially, theoretical studies, fundamentals and optics, such as

<https://www.nature.com/articles/s41377-020-00367-8>

<https://pubs.acs.org/doi/full/10.1021/acsp Photonics.1c00882>

<https://pubs.rsc.org/en/content/articlelanding/2021/nr/d1nr05805c>

Response: Thank you for the suggestion of the relevant and impactful references about plasmonic chirality. We have added the references as ref 42, 45, and 46.

Overall, I think this manuscript can be published in Nature Communications after these concerns being addressed.

We appreciate the reviewer for their time in assessing the manuscript and the constructive suggestions given. The revisions made based on these suggestions are made.

Reviewer #3 (Remarks to the Author):

This work is about enhancement of circularly polarized luminescence (CPL) in a complex molecular-plasmonic nano-system. The authors explored a mixed sensitizer-emitter luminescence upconversion micelle system, which on coating gold nanorods, produces enhanced luminescence and CPL. The enhanced luminescence in such systems was reported before, and the novelty in this work is the introduction of chirality in the molecules and observation of enhanced CPL. I would say that there is interest in finding ways to enhance CPL by plasmonic systems, as reported in this work. However, I feel that the publication of this work in Nature Communications would require more information and better mechanistic discussion, as detailed here:

1. There is not enough CPL data presented to understand what is going on in the studied system. The enhanced CPL spectra shown in Fig. 3A are fairly noisy for a supposedly strong CPL and much broader than the emission lines themselves, which makes one a bit suspicious about the measured CPL source, and whether there might be experimental artifacts in the measurements. In addition no g_{lum} spectra are shown, only peak values plotted in Fig. 4d. I suspect that the g_{lum} spectra would be also very noisy and they should be added, at least in the Supplementary Information, to understand if they have a physical meaning. Including the g_{lum} spectrum for the Au@Ag nanorod system, which is supposed to be the record one.

Response: We acknowledge the reviewer for the comment. To avoid the influence of the random results, we have measured the CPL spectra for several times and calculated the error bars for the corresponding g_{lum} values (Figure R5a). Moreover, the S-type emitters show a mirror-imaged CPL spectrum compared with the R-type emitters, indicating that the CPL signals are originated from the initial chirality rather than the artifacts (Figure R5b). As shown in Figure R5c-f, the g_{lum} spectra showed a stepwise increase from CPL via UC-CPL to PUC-CPL in a wide wavelength range. For the comparison of g_{lum} values, selecting the g_{lum} values around the peak values of emission is also a convenient and clear method in this community (*J. Phys. Chem. Lett.* 2014, 5, 316-321; *ACS Nano* 2019, 13, 3618-3628). As suggested by the reviewer, we have added the g_{lum} spectra in the current Supplementary information. Totally, we are confident that the plasmonic enhancement effect for CPL.

Figure R5. (a) Dissymmetry factor g_{lum} values of *R*-1IMP, *R*-1UCMP, and *R*-1UCMP/Au653 at peak values of emission. (b) CPL spectra of *R*-1IMP (black dot line) and *S*-1IMP (red dot line) in water excited by 400 nm. UC-CPL spectra of *R*-1UCMP (black dash line), *S*-1UCMP (red dash line), *R*-1UCMP/Au653 (black line) and *S*-1UCMP/Au653 (red line) in deaerated water excited by 635 nm laser. The emission intensities DC(V) were normalized to 0.5. Dissymmetry factor g_{lum} spectra of (c) *R*-/*S*-1IMP, (d) *R*-/*S*-1UCMP, (e) *R*-/*S*-1UCMP/Au653, and (f) *R*-1UCMP/Au812, *R*-1UCMP/Au737, *R*-1UCMP/Au@Ag659.

2. The explanation about connection of spin-polarization in the energy transfer process and the enhanced CPL is not clear to me. Eventually, the CPL magnitude (and g_{lum}) is determined by the properties of the emitting state alone. The above-mentioned explanation of the authors is valid only if the emitting state is spin-split and its sub-levels are selectively populated depending on the spin-state of the transferred electrons. Is this the case? Is there evidence for this? The CPL spectra are too noisy to decide whether there are spectral shifts between the two enantiomers, which might be the required evidence for the splitting in the emitting levels.

Response: We appreciate the reviewer for this insightful comment. We agree that there should be a spin split to explain the enhancement phenomenon. Unfortunately, because of the quite small energy difference, at ordinary temperatures the spectral shifts between two enantiomers are impossible to be observed from CPL spectra. However, some recent literatures have reported the spin split and magnet-controlled/induced CPL in organic systems (*Adv. Mater.* 2019, 31,1904857; *ChemPhotoChem* 2021,5, 969– 973). Thus, it is possible that electron spin polarization would influence the g_{lum} value of CPL. We have reorganized the part for mechanism explanation on page 10:

Addition to the well-known promotion of absorption and emission rates, we noted the improved excited-state absorption of *R*-1 in the micelle/plasmon hybrids. There were two possible sources of this phenomenon: (1) the enhanced absorption of sensitizers

led to an increasing generation rate of *R*-1 triplet excitons via TTET, (2) the improved absorption rate of triplet excited state induced by LSPR effect. Consequently, the total amount of triplet excitons was raised up. Among the chiral upconversion system, the quantitative balance between two spin orientations of triplet excitons was disturbed according to our previous report,²⁵ one kind of triplet excitons with certain spin orientation was more than its counterpart due to the chirality-induced electron spin polarization in TTET and TTA processes. This type of triplet excitons can be regarded as spin-polarized triplet excitons. In terms of AuNRs assisted UC-CPL process, the amount and percentage of spin-polarized triplet excitons should be increasing with the above-mentioned growth of triplet excitons. On the one hand, LSPR can enhance the chiral absorption which was based on electron transition, thus it was also probably to boost the electron spin polarization generated by the energy transfer that based on the electron exchange mechanism. Additionally, spin split and magnet-controlled/induced CPL have been reported in pure organic systems.^{61,62} These phenomena suggested a possibility for electron spin polarization to have an impact on chiral emission. Therefore, we proposed that the effect of LSPR-enhanced chirality-induced spin polarization played an important role in amplifying the g_{lum} values of UC-CPL. As shown in Fig. 4e, the chirality-induced spin polarization both in TTET and TTA has been enhanced due to the coupling with plasmon nanorods. Consequently, the UC-CPL resulted from spin-polarized singlet excitons showed a higher g_{lum} value.

3. Is it possible that changes in CPL on attaching the micelles to the nanorods occur due to reorganization of the molecules in the micelles? I think that the authors should discuss this reasonable possibility and try to bring evidence for/against it.

Response: If the changes of CPL mostly originated from the reorganization of the chiral emitters, it should always show obvious changes on CPL no matter there is a LSPR electromagnetic field or not. We measured the downshifting CPL to explore the mechanism according to the reviewer's suggestions. The downshifting CPL excited by 400 nm and the electromagnetic fields of the nanorods are very weak under this excitation wavelength. If the reorganization of the chiral emitters is the main reason for enhancement, the downshifting CPL should also be quite different after attaching the micelles to the nanorods. However, there are only slight increase (less than twice for enhancement) in the magnitude of downshifting CPL. Therefore, the main reason for the amplification of CPL is LSPR enhancement effect.

4. The dynamics of the excited states is less interesting here. The fact that lifetimes are shortened by the presence of metal nanostructures is well known and I think that it does not contribute much to the understanding of CPL effects. I would reduce this part and improve discussion of the above three points.

Response: We agree that the shortening of lifetime is well known in plasmon-coupling luminescent systems. This phenomenon is more studied to prove the increase of radiative decay rate of emitters. We have simplified the discussion of time-resolved

photo upconversion on page 8 and revised the mechanism for CPL enhancement that has been shown above.

On page 9,

Additionally, time-resolved photoluminescence spectra revealed that the lifetime of *R*-1UCMP showed an obvious decrease from 102 to 82 μ s in the presence of Au653 (Fig. 3f), which was resulted from the LSPR-caused faster emission rates,^{49,57} further demonstrating the coupling of LSPR to *R*-1/PdTPBP pair.

After addressing the above issues the work could be considered again for publication.

We acknowledge the reviewer for these insightful comments and have made revision according to the suggestions.

Reviewer #4 (Remarks to the Author):

The manuscript by Duan et al. reports an upconverting system showing CPL. g_{lum} factors are significantly enhanced upon adsorption to metal surface, such as Au rods. The main and most novel point of the work is that plasmon resonance can amplify CPL associated to TTA-UC (triplet-triplet annihilation-upconversion). Given the interest of a large community on these topics, I think that the manuscript could be accepted after careful revisions.

We thank the reviewer for their time in assessing the manuscript and the constructive suggestions given. The revisions made based on these suggestions are detailed below.

1) It is not entirely clear to me how plasmon resonance works to amplify TTA-UC CPL. What is the effect of plasmon resonance on regular (down-converted) CPL? It would be useful to see a full comparison between regular and UC CPL for the complete assembly (system/Au). Moreover, Why the presence of Au nanoparticles has only little effects on the CD properties? This is an important point, since the authors claim that "either ground- or excited-state chirality can be enhanced by plasmonic LSPR effect".

Response: Thank you for this constructive suggestion given. We have demonstrated that the local electromagnetic field of plasmon nanorods plays a key role for the enhancement by employing different types nanorods to the hybrid systems, such as Au653, Au737, Au812, and Au@Ag659. Thus, the main reason for the weak effects on the CD properties is that the extinction band of *R-1* is not well matching the LSPR wavelength of Au653. This idea can also be demonstrated by the measurement of downshifting CPL. As shown in Fig. R5, under excitation with 400 nm, the g_{lum} values of *R-1MP/Au653* only exhibit slight increase compared with *R-1MP*, and even at high concentration of Au653, the scattering of Au653 make the g_{lum} values fluctuant. Because the local electromagnetic field of Au653 is pretty weak at such excitation wavelength. We have added the content about this result in the revised manuscript and Supplementary information.

On page 7 in the revised manuscript:

We noted that the amplification level of CD was much smaller than that of UC-CPL. This may be explained by the extinction band of *R-1* was not well matching the LSPR wavelength of Au653. Additionally, the measurement of regular downshifting CPL was carried out in *R-1MP/Au653* composites. Different with the UC-CPL, upon excitation with 400 nm, the *R-1* mixed with different concentrations of Au653 only showed slight increase on g_{lum} values (Supplementary Fig. 9). This result indicated that although the emission band of *R-1* overlapped with the transverse LSPR of Au653, the electromagnetic field of Au653 was too weak to allow a dramatic enhancement effect at such excitation wavelength. This behavior will be further investigated in the following experimentation by employing plasmonic nanorods with various longitudinal LSPR wavelength.

Figure R5. (a) CPL spectra and (b) g_{lum} values of *R*-1MP and *R*-1MP/Au653 composites in water. $[R-1] = 5 \times 10^{-5} \text{ mol L}^{-1}$, $[CTAB] = 10^{-2} \text{ mol L}^{-1}$, $\lambda_{ex} = 400 \text{ nm}$.

2) “Chirality-induced spin polarization”, please clarify.

Response: Thank you for your suggestion. Chirality-induced spin polarization is an unequal population distribution of electron/charge that caused by molecular chirality (*Annu. Rev. Phys. Chem.* 2015, 66, 263-281; *Proc. Natl. Acad. Sci. U. S. A.* 2017, 114, 2474-2478). In the chiral upconversion system, the chirality-induced spin polarization can cause the unbalanced amount of triplet excitons with certain spin orientations generated through TTET and TTA. This type of triplets is called as spin-polarized triplet excitons. We have clarified this effect. The mechanism for plasmon enhanced UC-CPL is also rewrote on page 10, it reads:

Addition to the well-known promotion of absorption and emission rates, we noted the improved excited-state absorption of *R*-1 in the micelle/plasmon hybrids. There were two possible sources of this phenomenon: (1) the enhanced absorption of sensitizers led to an increasing generation rate of *R*-1 triplet excitons via TTET, (2) the improved absorption rate of triplet excited state induced by LSPR effect. Consequently, the total amount of triplet excitons was raised up. Among the chiral upconversion system, the quantitative balance between two spin orientations of triplet excitons was disturbed according to our previous report,²⁵ one kind of triplet excitons with certain spin orientation was more than its counterpart due to the chirality-induced electron spin polarization in TTET and TTA processes. This type of triplet excitons can be regarded as spin-polarized triplet excitons. In terms of AuNRs assisted UC-CPL process, the amount and percentage of spin-polarized triplet excitons should be increasing with the above-mentioned growth of triplet excitons. On the one hand, LSPR can enhance the chiral absorption which was based on electron transition, thus it was also probably to boost the electron spin polarization generated by the energy transfer that based on the electron exchange mechanism. Additionally, spin split and magnet-controlled/induced CPL have been reported in pure organic systems.^{61,62} These phenomena suggested a possibility for electron spin polarization to have an impact on chiral emission. Therefore, we proposed that the effect of LSPR-enhanced chirality-induced spin

polarization played an important role in amplifying the g_{lum} values of UC-CPL. As shown in Fig. 4e, the chirality-induced spin polarization both in TTET and TTA has been enhanced due to the coupling with plasmon nanorods. Consequently, the UC-CPL resulted from spin-polarized singlet excitons showed a higher g_{lum} value.

3) “The upconverted luminescence intensity versus the concentration of metal nanoparticles showed a volcano shape” please clarify.

Response: We apologize for this mistake. We have corrected this sentence on page 5 reads:

By increasing Au653 amount, the upconverted luminescence intensity versus the concentration of metal nanoparticles showed a volcano-shape curve (Fig. 2e).

The volcano-shape curve describes the trend of upconverted luminescence intensity that firstly increases but decreases after getting a maximum value with the change of the concentration of metal nanoparticles. This way of expression is commonly used in literatures: *Electrochimica Acta* 2002, 47, 3723-3732; *Faraday Discuss.*, 2021, 229, 62-74.

4) In Fig. 3 and elsewhere, excitation wavelengths should be reported.

Response: Thanks for your reminder. We have checked all places that excitation wavelengths should be presented.

5) The free micelle concentration should be estimated to have an accurate value for g factors/enhancement.

Response: We agree that the estimation of real enhancement factor is important. We have attempted to remove the free micelles by centrifugation and obtained the ratio of them by comparing the integrating extinction intensity between R -1UCMP and supernatant of R -1UCMP/Au653 (Fig. R6). The calibrated enhancement values are 3.3 and 15.7 times for upconverted emission and g_{lum} value, respectively. We admit that this method is not perfect to distinguish the free micelle, but sufficient to roughly estimate the enhancement factor. We have described this experimentation on page 7:

The true enhancement factor per attached micelle should be larger because a fraction of the micelle particles was free as exhibited in the inserted TEM image of R -1UCMP/Au653 (Fig. 2c). Here, we attempted to obtain the real enhancement factor through comparing the integrating extinction intensities of R -1UCMP and the supernatant of R -1UCMP/Au653 after removal of Au653 by centrifugation. In this operation, the R -1UCMP/Au653 composite with a $\text{mol}_{653}/\text{mol}_{R-1}$ at $3.0 \times 10^{-6}/1$ was selected due to high-concentration Au653 can quench the upconverted emission. As shown in Supplementary Fig. 7, the extinction intensity of supernatant was 58% of the

pure *R*-1UCMP, which meant that approximately 42% of the micelle particles combined with the Au653. Accordingly, the corrected enhancement factors were 3.3 and 15.7 times for upconverted emission and g_{lum} value, respectively. With the deviations in the centrifugation process this method was not perfect, but sufficient to roughly estimate the enhancement factor in a very simple and convenient fashion.

Figure R6. Extinction spectra of *R*-1UCMP and supernatant of *R*-1UCMP/Au653 after centrifugation. $[R-1] = 5 \times 10^{-5} \text{ mol L}^{-1}$, $[\text{PdTPBP}] = 10^{-5} \text{ mol L}^{-1}$, $[\text{CTAB}] = 10^{-2} \text{ mol L}^{-1}$, $\text{mol}_{\text{Au653}}/\text{mol}_{R-1} = 3.0 \times 10^{-6}/1$.

6) “Due to the competed absorption”, it should read “competing”.

Response: This sentence has been revised on page 5: upconverted luminescence intensity would decrease due to the **competing** absorption at 635 nm between sensitizer and Au653 and the reabsorption of upconverted luminescence by Au653.

7) “Indicating that the formation of micelle remained the chirality of chiral emitters”, please check this sentence.

Response: Thanks for the reviewer’s reminding. We have corrected this part to: On page 12, There are no obvious changes in the corresponding CD spectra of *R*-/*S*-2MP as well as in CPL spectra (Supplementary Fig. S17), verifying the optical activities of *R*-/*S*-2MP.

8) The manuscript should be checked throughout for grammar mistakes and to improve clarity in general

Response: We appreciate the close reading and constructive criticism of the manuscript

from the reviewer. Based on these comments we are happy to resubmit a strengthened manuscript. We have edited for spelling and clarity throughout again also.

Reviewer comments, second round

Reviewer #1 (Remarks to the Author):

The reviewer has carefully read the authors' responses to the reviewer's comments and the revised manuscript. In particular, the reviewer finds the discussion of the enhancement of CPL activity noteworthy, and believes that the authors have presented the best possible experimental data. Therefore, we believe that this manuscript is worthy of publication in Nature Communications.

Reviewer #2 (Remarks to the Author):

In this paper, the authors proposed plasmonic nanoparticles-integrated chiral nanocomposites to enhance the upconverted PL and circularly polarized luminescence. The minor concerns are well-revised. The author provided a detailed explanation about why the Ag core-shell nanoparticle exhibits stronger plasmonic resonance compared with the nanoparticles only composed of Au. In addition, the new energy-level diagram is added in Figure 4e. This diagram shows the plasmon-assisted UC-CPL process of R-1/PdTPBP and I think this can help the readers to understand the upconversion-CPL process. Overall, the manuscript is improved significantly so I would recommend the paper be published in Nature communications.

Reviewer #3 (Remarks to the Author):

I have read the revised paper and the response of the authors to my comments, as well as other reviewers' comments.

I still have a problem with my two main concerns:

1. The nature of the CPL signals - the glum spectra are extremely broad and flat in most of the spectral range. Hence, their use is questional, regarding quantification of the magnitude of enhancement of the CPL effect. I think that the authors should seriously address this problem and think about possible ways to improve on that, or altenatively put the quantitative assesment of the magnitude of CPL enhancement in question.
2. The use of spin-polarized exciton levels as the source of enhanced CPL is still highly speculative. The added paragraph about this still does not contain any experimental proof. Hence, if such a high enhancement really exists (in view of my first comment) than the authors should be careful about this speculation and present it as such.

I therefore think that publication in Nat. Comm. is still not appropriate until the concern about the quantification of enhanced CPL is met.

additional small comments:

1. The english still requires serious improvement.
2. There is a missing inset in fig. 2c - a TEM image of the nanorods with the micelles attached, I believe.
3. What is the quantum yield of the R-1UCMP/Au653 system? This could help deducing on the source of luminescence enhancement in this hybrid system relative to the R-1UCMP alone.
4. Fig. 3f - the horizontal scale should be ps, right?

Reviewer #4 (Remarks to the Author):

The authors have revised the manuscript satisfactorily. I think that some points of the mechanism proposed by the authors still need a more careful explanation and possibly more experimental evidence, but this can be postponed to a later more specialized publication. So I recommend publication of the manuscript as is.

Reviewer #1 (Remarks to the Author):

The reviewer has carefully read the authors' responses to the reviewer's comments and the revised manuscript. In particular, the reviewer finds the discussion of the enhancement of CPL activity noteworthy, and believes that the authors have presented the best possible experimental data. Therefore, we believe that this manuscript is worthy of publication in Nature Communications.

We thank the reviewer for the endorsement and the insightful comments given.

Reviewer #2 (Remarks to the Author):

In this paper, the authors proposed plasmonic nanoparticles-integrated chiral nanocomposites to enhance the upconverted PL and circularly polarized luminescence. The minor concerns are well-revised. The author provided a detailed explanation about why the Ag core-shell nanoparticle exhibits stronger plasmonic resonance compared with the nanoparticles only composed of Au. In addition, the new energy-level diagram is added in Figure 4e. This diagram shows the plasmon-assisted UC-CPL process of R-1/PdTPBP and I think this can help the readers to understand the upconversion-CPL process. Overall, the manuscript is improved significantly so I would recommend the paper be published in Nature communications.

We appreciate the reviewer for their recognition of our work and the great help in the improvement of the manuscript.

Reviewer #3 (Remarks to the Author):

I have read the revised paper and the response of the authors to my comments, as well as other reviewers' comments.

I still have a problem with my two main concerns:

1. The nature of the CPL signals - the g_{lum} spectra are extremely broad and flat in most of the spectral range. Hence, their use is questionable, regarding quantification of the magnitude of enhancement of the CPL effect. I think that the authors should seriously address this problem and think about possible ways to improve on that, or alternatively put the quantitative assessment of the magnitude of CPL enhancement in question.

Response: The CPL originates from the initial chirality of emitters and is a phenomenon that relates to the luminescence. Therefore, in organic system, the CPL spectra are broad like luminescence spectra. The g_{lum} spectra are calculated from CPL spectra, so it is not surprising that the g_{lum} spectra are also broad. In addition, the g_{lum} spectra with flat shape are normal because the CPL usually emits from single excited-state chirality, many literatures have reported the flat spectra regardless of whether the CPL spectra were carried out on commercialized or home-built instruments: *J. Phys. Chem. C* 2018, 122, 24924–24932; *Research* 2020, 2020, 3839160; *Chem. - Eur. J.* 2014, 20, 8386–8390; *Chem. Sci.* 2012, 3, 2737–2747; *J. Phys. Chem. Lett.* 2014, 5, 316–321. As for the estimation of CPL enhancement, it is widely used to select the g_{lum} value corresponding to the emission peak for comparison (*Chem. -Eur. J.*, 2013, 19, 14090–14097; *Chem. -Eur. J.* 2016, 22, 12910; *Angew. Chem. Int. Ed.* 2021, 60, 222; *Chem. Sci.* 2019, 10, 1294–1301), and this method is approved in this community. We hope this assessment method could be acceptable for the reviewer.

2. The use of spin-polarized exciton levels as the source of enhanced CPL is still highly speculative. The added paragraph about this still does not contain any experimental proof. Hence, if such a high enhancement really exists (in view of my first comment) than the authors should be careful about this speculation and present it as such.

Response: We agree that more study would be useful to understand mechanism of enhancement. As triplet-triplet annihilation is significantly affected by the relative density of triplet excitons with opposite spin orientation, we mainly studied the TTA efficiencies of *R*-1UCMP and *R*-1UCMP/Au653 systems. As shown in Figure R1, *R*-1UCMP/Au653 shows a lower TTA efficiency than that of *R*-1UCMP, indicating the triplet excitons with opposite spin orientation are more unequal in *R*-1UCMP/Au653. This result means probably the plasmon-assisted chiral upconversion possesses higher exciton spin polarization level. We have added this data and description in the revised manuscript and supplementary information. The revision on page 10 reads:

It has been proven that the enhancement of chiroptical properties originated from the coupling effect between chiral substances and plasmonic localized electromagnetic field.^{52,59,60} However, in a UC-CPL system, one should take account of multiple photoinduced processes. In addition to the well-known promotion of absorption and

emission rates, we noted the improved excited-state absorption of *R*-1 in the micelle/plasmon hybrids. There were two possible sources of this phenomenon: (1) the enhanced absorption of sensitizers led to an increasing generation rate of *R*-1 triplet excitons via TTET, (2) the improved absorption rate of triplet excited state induced by LSPR effect. Consequently, the total amount of triplet excitons was raised up. Among the chiral upconversion system, the quantitative balance between two spin orientations of triplet excitons was disturbed according to our previous report,²⁵ one kind of triplet excitons with certain spin orientation was more than its counterpart due to the chirality-induced electron spin polarization in TTET and TTA processes. This type of triplet excitons can be regarded as spin-polarized triplet excitons. In terms of AuNRs assisted UC-CPL process, the amount and fraction of spin-polarized excitons should increase with the increase of total amount of triplet excitons. To verify our hypothesize, we evaluated the the intensity of spin polarization by comparing TTA efficiencies between *R*-1UCMP and *R*-1UCMP/Au653, because the TTA efficiency (Φ_{TTA}) is significantly affected by the relative density of triplet excitons with opposite spin orientation. Unfortunately, the calculated Φ_{TTA} of both *R*-1UCMP and *R*-1UCMP/Au653 were approximately unit based on the upconversion decay shown in Fig. 3f, because the density of triplet excitons was saturated under high-power excitation. Therefore, we turned to estimate Φ_{TTA} upon low-power excitation. Interestingly, the Φ_{TTA} of *R*-1UCMP/Au653 (58 %) was lower than that of *R*-1UCMP (67 %, Supplementary Figure 16), suggesting that the density of triplet excitons with opposite spin orientation was more unbalanced in plasmon-assisted systems. This result agreed with our hypothesize that *R*-1UCMP/Au653 probably had higher spin polarization level. On the one hand, LSPR can enhance the chiral absorption which was based on electron transition; thus, it was also probably to boost the electron spin polarization generated by the energy transfer that based on the electron exchange mechanism. Additionally, spin split and magnet-controlled/induced CPL have been reported in pure organic systems.^{61,62} These phenomena suggested a possibility for electron spin polarization to have an impact on chiral emission. Therefore, we propose that the effect of LSPR-enhanced chirality-induced spin polarization is critical to amplifying the g_{lum} values of UC-CPL. As shown in Fig. 4e, the chirality-induced spin polarization both in TTET and TTA has been enhanced due to the coupling with plasmon nanorods. Consequently, the UC-CPL resulted from spin-polarized singlet excitons showed a larger g_{lum} value.

Figure R1. The fit of Φ_{TTA} with Equation S2 (red lines) accorded to UC-PL decay of (a) *R*-1UCMP and (b) *R*-1UCMP/Au653 at 476 nm. $[R-1] = 5 \times 10^{-5} \text{ mol L}^{-1}$, $[\text{PdTPBP}] = 10^{-5} \text{ mol L}^{-1}$, $[\text{CTAB}] = 10^{-2} \text{ mol L}^{-1}$, $\text{mol}_{\text{Au653}}/\text{mol}_{R-1} = 3 \times 10^{-6}/1$. Excitation source: 635 nm laser with power of 675 mW cm^{-2} .

I therefore think that publication in Nat. Comm. is still not appropriate until the concern about the quantification of enhanced CPL is met.

Response: We thank the reviewer for the comments. We indeed are thinking about using circularly polarized ultrafast spectroscopy to determine the chiral behavior in excited state, which is useful to deeply understand the enhancement mechanism. But this study is beyond the scope of this current work which focuses on plasmon-enhanced UC-CPL. We'll work on it in the future. We hope the revised manuscript has met the reviewer's requirements.

additional small comments:

1. The English still requires serious improvement.

Response: Thanks for your suggestion. We have worked on the manuscript for a long time and asked a native English speaker to assist us with correcting the spelling, grammar, word use, and punctuation throughout our manuscript. The changes in the manuscript are highlighted in yellow mark.

2. There is a missing inset in fig. 2c - a TEM image of the nanorods with the micelles attached, I believe.

Response: Thanks for the reminding. The TEM image has been added in the supplementary information as Supplementary Figure 4b. We cited this figure on page 7 in the revised manuscript:

The true enhancement factor per attached micelle should be larger because a fraction of the micelle particles was free as exhibited in the inserted TEM image of *R*-1UCMP/Au653 (Supplementary Fig. 4b).

3. What is the quantum yield of the *R*-1UCMP/Au653 system? This could help deducing on the source of luminescence enhancement in this hybrid system relative to the *R*-1UCMP alone.

Response: We have measured the quantum yield of *R*-1UCMP and *R*-1UCMP/Au653 by a relative method (Figure R2). Indeed, the *R*-1UCMP/Au653 possesses higher efficiency than *R*-1UCMP, confirming the plasmon-induced upconversion enhancement. We have added the description and data in the revised manuscript and supplementary information.

On page 5,

The upconversion quantum yield of *R*-1UCMP increased from 0.2% to 0.5% after mixing with Au653 (Supplementary Fig. 5e).

Figure R2. UC quantum yield Φ_{UC} of *R*-1UCMP/Au653 and *R*-1UCMP in aqueous solution as a function of excitation intensity of the 635 nm laser.

4.Fig. 3f - the horizontal scale should be ps, right?

Response: We guess the reviewer thinks Fig. 3f shows the transient absorption spectrum. However, Fig. 3f is the upconversion decay spectrum. So the horizontal axial with a μs scale is right.

Reviewer #4 (Remarks to the Author):

The authors have revised the manuscript satisfactorily. I think that some points of the mechanism proposed by the authors still need a more careful explanation and possibly more experimental evidence, but this can be postponed to a later more specialized publication. So I recommend publication of the manuscript as is.

We thank the reviewer for their time in reassessing the manuscript and the approval of our work. We will try our best to further explore the mechanism in the future.